# Hydrogen spillover assisted by oxygenate molecules over nonreducible oxides

Mingwu Tan[1], Yanling Yang[1], Ying Yang[1], Jiali Chen[1], Zhaoxia Zhang[1], Gang Fu [1], Jingdong Lin [1], Shaolong Wan[1], Shuai Wang [1✉] & Yong Wang [2✉]

Spontaneous migration of atomic hydrogen species from metal particles to the surface of their support, known as hydrogen spillover, has been claimed to play a major role in catalytic processes involving hydrogen. While this phenomenon is well established on reducible oxide supports, its realization on much more commonly used non-reducible oxides is still challenged. Here we present a general strategy to enable effective hydrogen spillover over non-reducible $SiO_2$ with aid of gaseous organic molecules containing a carbonyl group. By using hierarchically-porous-$SiO_2$-supported bimetallic Pt-Fe catalysts with Pt nanoparticles exclusively deposited into the micropores, we demonstrate that activated hydrogen species generated on the Pt sites within the micropores can be readily transported by these oxygenate molecules to Fe sites located in macropores, leading to significantly accelerated hydrodeoxygenation rates on the latter sites. This finding provides a molecule-assisted approach to the rational design and optimization of multifunctional heterogeneous catalysts, reminiscent of the role of molecular coenzymes in bio-catalysis.

[1] State Key Laboratory for Physical Chemistry of Solid Surfaces, Collaborative Innovation Center of Chemistry for Energy Materials, National Engineering Laboratory for Green Chemical Productions of Alcohols-Ethers-Esters, and College of Chemistry and Chemical Engineering, Xiamen University, Xiamen 361005, China. [2] Voiland School of Chemical Engineering and Bioengineering, Washington State University, Pullman, WA 99164, USA. ✉email: shuaiwang@xmu.edu.cn; yongwang@pnnl.gov

Hydrogen spillover has been coined to denote an important effect involved in heterogeneous catalysis[1,2], in which atomic H species produced from $H_2$ dissociative adsorption on a catalytic site spontaneously migrate to a different type of site that cannot readily dissociate $H_2$ molecules under the same condition[3–5]. This effect not only accounts for many phenomena observed in hydrogenation/hydrogenolysis processes[6,7], but also makes it possible to decouple the catalytic functions between $H_2$ dissociation (to form active H atoms) and subsequent hydrogenation with these atomic H species, rendering the control of these catalytic sequences separately and precisely towards more efficient synthesis of chemicals[8–14].

Extensive studies have been devoted to investigating the mechanism of hydrogen spillover on oxide supports[3–6]. On one hand, it is commonly accepted that atomic H species can migrate along surfaces of reducible oxide (e.g., $TiO_2$ and $WO_3$) via coherent proton-electron movements[15,16], in which the H atom gives its electron to a reducible metal cation of the oxide support and diffuses in the form of proton. On the other hand, the possibility of hydrogen spillover on a nonreducible oxide (e.g., $SiO_2$ and $Al_2O_3$) has been debated for a long time[4,17], and only until recently it was found by combined experimental and theoretical evidences that the mobility of H atoms on a nonreducible $Al_2O_3$ support is orders of magnitude slower and restricted to a much shorter distance (less than sub-nanometer) than the case for a reducible $TiO_2$ support[5,18]. Although some studies suggest that the presence of sufficient acidic surface OH groups or introducing defect sites into the oxide surface may promote the hydrogen transfer process[9–11], a general and robust strategy is still highly desired for achieving an effective hydrogen spillover on nonreducible oxides with a low density of acid sites or defects, which are widely used as catalyst supports in industry because of their excellent thermochemical stability, low cost, and tunable acid-base property.

In our previous work[19], Pt-Fe bimetallic catalysts (denoted as Pt@-Fe@SiO$_2$) with Pt and Fe nanoparticles encapsulated in the micropores and macropores of a hierarchically structured $SiO_2$ support, respectively, showed activity for hydrodeoxygenation (HDO) of pyrolysis bio-oil vapor on Fe sites with $H_2$ generated from steam reforming on Pt sites. However, questions such as how the H species are stabilized and how the H species "shuttle" or communicate between the spatially isolated Pt and Fe sites remained unanswered. Here, using HDO of lignin-derived guaiacol to produce value-added aromatic hydrocarbon products (e.g., benzene, toluene, and xylene, denoted collectively as BTX) as a model reaction, we unveil that an efficient hydrogen spillover from the Pt sites to the Fe sites occurs over the $SiO_2$ support with aid of gaseous oxygenate molecules containing a carbonyl functional group such as aldehydes, ketones, and esters (illustrated in Fig. 1), which leads to significantly accelerated HDO rates and is reminiscent of molecular coenzymes that assist in enzyme activity in a similar manner[20].

## Results

### Synthesis and characterization of Pt@-Fe@SiO$_2$ catalysts.
As reported in our previous study[18], Pt@-Fe@SiO$_2$ catalysts were synthesized via a dual template method with a nonionic polyglycol ether surfactant (Tergitol 15-S-5) and a cellulose extraction thimble used as the templates to generate micropores and macropores in $SiO_2$, respectively (described in Supplementary Fig. 1 of the supporting information). The content of Pt in these catalysts varied within 0.5–2.0 wt%, while the content of Fe was kept at 10 wt%. These catalysts are denoted as $x$Pt@-10Fe@SiO$_2$ ($x = 0.5$–2.0) henceforth. For comparison, 1Pt@SiO$_2$ with 1.0 wt% Pt encapsulated within the micropores of $SiO_2$ and 10Fe@SiO$_2$

with 10 wt% Fe mainly deposited within the macropores of $SiO_2$ were also synthesized using this dual template method.

The micropores of the synthesized $SiO_2$ support possess a size distribution mainly within 1.0–2.0 nm as determined by $N_2$ physisorption isotherms (Supplementary Fig. 2a) and small angle X-ray powder diffraction (XRD) spectra (Supplementary Fig. 3a), and show a well-ordered wormhole-like feature according to transmission electron microscopy (TEM) images (Supplementary Fig. 4). Mercury intrusion porosimetry measurement further confirms the existence of macropores with a diameter ranging from 100 to 160 nm for the $SiO_2$ support (Supplementary Fig. 2b). This hierarchically porous structure is well maintained after depositing the Pt and Fe nanoparticles onto the $SiO_2$ support, consistent with insignificant changes of the $N_2$ physisorption isotherms and average pore sizes (Supplementary Fig. 5).

XRD patterns of these Pt@-Fe@SiO$_2$ samples show the diffraction peaks belonging to cubic Fe crystalline phases (e.g., $2\theta = 44.6°$), whereas characteristic diffraction peaks associated with the Pt nanoparticles are undetectable reflective of their high dispersion (Supplementary Fig. 6). TEM images confirm uniformly dispersed Pt nanoparticles of around $1.6 \pm 0.3$ nm for the Pt@-Fe@SiO$_2$ catalysts, irrespective of the presence of Fe species (Fig. 2a and b) and the Pt loading amount within 0.5–2.0 wt% (Supplementary Fig. 7), which reflects the confinement effect on the growth of these Pt nanoparticles within micropores. In contrast, the Fe nanoparticles deposited in the macropores show much larger sizes (e.g., $6.5 \pm 0.7$ nm for 1Pt@-10Fe@SiO$_2$; $7.2 \pm 0.7$ nm for 10Fe@SiO$_2$, Fig. 2a, c). $H_2$-temperature-programmed reduction ($H_2$-TPR) profiles of the oxide precursors for 10Fe@SiO$_2$ and 1Pt@-10Fe@SiO$_2$ show nearly identical temperatures (c.a. 441 °C) for the reduction of $Fe_2O_3$ (Fig. 2d), consistent with the spatial isolation between Pt and Fe nanoparticles in these Pt@-Fe@SiO$_2$ catalysts. Otherwise, the promoted reduction of $Fe_2O_3$ species is expected if they are in contact with Pt nanoparticles as found for the Pt-Fe/SiO$_2$ catalyst (1 wt% Pt, 10 wt% Fe) synthesized via a conventional co-impregnation method, which exhibits the $H_2$-TPR peak at an apparently lower temperature (419 °C, Fig. 2d). Moreover, these close reduction temperatures for the $Fe_2O_3$ species also confirm that hydrogen spillover is inhibited on the $SiO_2$ support used in this study.

### Rates and selectivity of guaiacol hydrodeoxygenation on Pt@-Fe@SiO$_2$.
Catalytic performances of 1Pt@SiO$_2$, 10Fe@SiO$_2$, and 1Pt@-10Fe@SiO$_2$ were compared in guaiacol HDO (450 °C, 0.5 kPa guaiacol, 50 kPa $H_2$, 4.0 mL s$^{-1}$ g$_{cat}^{-1}$ of space velocity), which has been widely applied as a model reaction for studying the upgrading of lignin-derived phenolics to aromatic hydrocarbons[21–23]. As illustrated in Fig. 3, the C–O bond cleavage in guaiacol on a metal catalyst can lead to sequential formations of catechol, phenol, and benzene. Concomitant with these conversions, the methyl species generated in the catechol formation can recombine with phenol to yield anisole and cresol, which are grouped with phenol as the partially deoxygenated products (denoted here as PAC), while the corresponding methylated derivatives of benzene mainly include toluene and xylene and are grouped with benzene since they are all fully deoxygenated products (i.e., BTX). Alternatively, guaiacol, catechol, and other phenolics can undergo hydrogenation of their aromatic ring on a metal catalyst, followed by subsequent decomposition into $C_1$ and $C_2$ gaseous molecules (e.g., methane, CO, $CO_2$, and ethane) via C–C bond cleavage. Previous studies have shown that the selectivity of guaiacol HDO is sensitive to the nature of the metal catalyst[24]. In particular, Fe is highly selective for the C–O bond cleavage[25–27], while those C=C hydrogenation

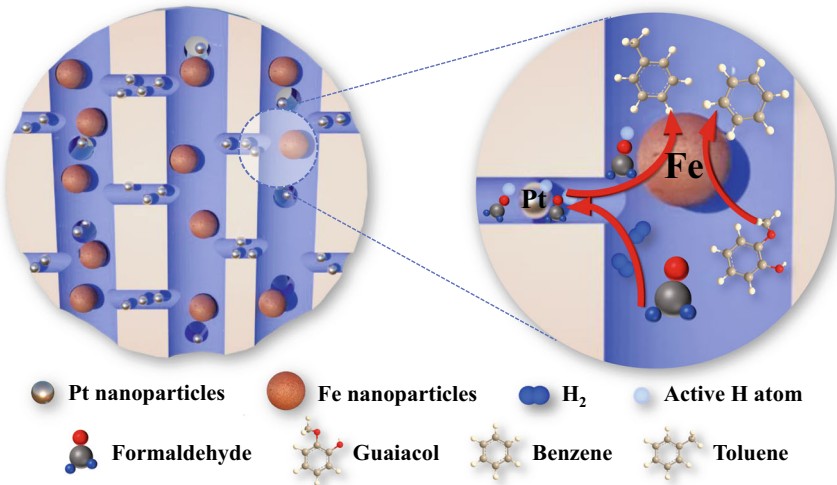

**Fig. 1 Hydrogen spillover enhanced by oxygenate additives during catalysis.** An oxygenate molecule (e.g., formaldehyde) acts as a H-carrier to promote hydrogen spillover for guaiacol hydrodeoxygenation on a hierarchically porous Pt@-Fe@SiO$_2$ catalyst.

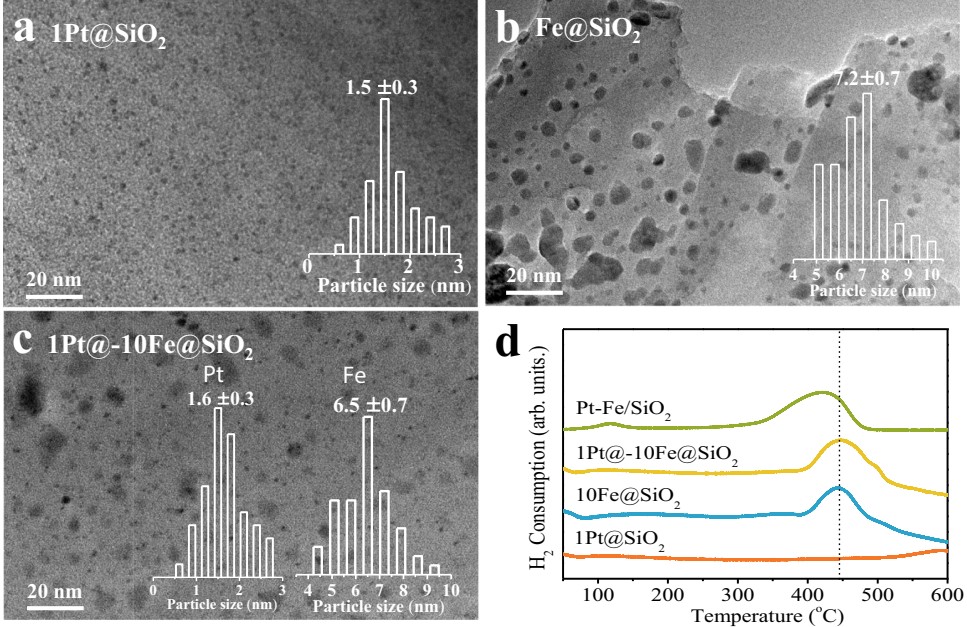

**Fig. 2 Structural characterization of Pt@-Fe@SiO$_2$ catalysts.** TEM images for (**a**) 1Pt@SiO$_2$; (**b**) 10Fe@SiO$_2$, and (**c**) 1Pt@-10Fe@SiO$_2$; (**d**) H$_2$-TPR profiles for the oxide precursors of these samples with a Pt-Fe/SiO$_2$ catalyst (1 wt% Pt, 10 wt% Fe) synthesized via a co-impregnation method as reference.

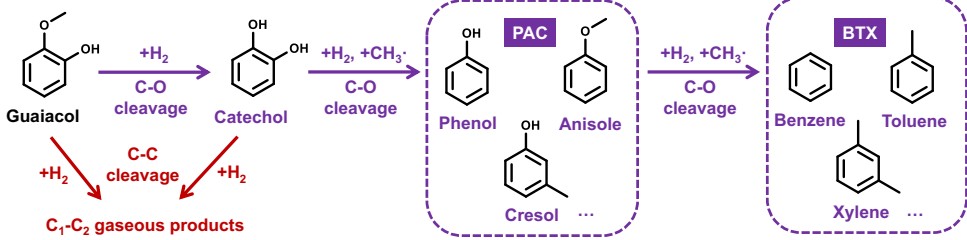

**Fig. 3 Reaction pathways of guaiacol hydrodeoxygenation.** Only relevant products observed in this study are included here.

and C–C hydrogenolysis reactions are favored on noble metals (e.g., Pt and Pd)[20,21].

1Pt@SiO$_2$ showed a low guaiacol conversion of 15.8% with catechol and PAC as the predominant products, the selectivities of which were 70.2% and 17.9%, respectively (Entry 1, Table 1).

The main activity of 1Pt@SiO$_2$ is likely contributed by the residual active sites on the SiO$_2$ support, because the guaiacol conversion (11.2%) and the combined selectivity of catechol and PAC (84.3%) obtained on the parent SiO$_2$ support (Entry 2, Table 1) were close to those of 1Pt@SiO$_2$. In contrast, a referenced

**Table 1 Catalytic performance of gas-phase guaiacol hydrodeoxygenation[a].**

| Entry | Catalyst | Guaiacol conversion (%) | Carbon selectivity (%) | | | | Formation rate of BTX (mol $mol_{Fe/Pt}^{-1}$ $h^{-1}$) |
|---|---|---|---|---|---|---|---|
| | | | Catechol | PAC[b] | BTX[c] | $C_1$-$C_2$[d] | |
| 1 | 1Pt@SiO$_2$ | 15.8 | 17.9 | 70.2 | 3.5 | <0.1 | 0.01 |
| 2 | SiO$_2$ | 11.2 | 33.1 | 51.2 | <0.1 | <0.1 | – |
| 3 | 3 wt% Pt/C | 97.4 | 0.5 | 18.9 | 5.3 | 71.9 | 0.09 |
| 4 | 10Fe@SiO$_2$ | 94.8 | 5.1 | 46.4 | 48.5 | <0.1 | 0.83 |
| 5 | 1Pt@-10Fe@SiO$_2$ | 96.3 | 2.2 | 48.1 | 48.8 | <0.1 | 0.84 |

[a]Reaction condition: 450 °C, 0.5 kPa guaiacol, 50 kPa H$_2$, balanced by N$_2$, 0.25 g•s•mL$^{-1}$ space velocity.
[b]Denoted for phenol, anisole, and cresol.
[c]Denoted for benzene, toluene, and xylene.
[d]Mainly including methane, CO, CO$_2$, and ethane.

non-encapsulated Pt/C catalyst (3 wt% Pt) exhibited a nearly complete guaiacol conversion (97.4%, Entry 3 in Table 1), in which the $C_1$ and $C_2$ gaseous products (71.9% selectivity) were formed from the successive C–C/C–O hydrogenolysis reactions prevailing among the products. The above data clearly indicate that most of the Pt nanoparticles on 1Pt@SiO$_2$ are not accessible to the bulky guaiacol reactants due to their successful encapsulation within the micropores of SiO$_2$.

Different from the above catalysts, a guaiacol conversion of 94.8% with selectivities to PAC and BTX of 46.4% and 48.5%, respectively, was achieved on 10Fe@SiO$_2$ (Entry 4, Table 1). As expected, guaiacol can readily adsorb on the Fe active sites because of the preferential deposition of the Fe nanoparticles in the macropores of SiO$_2$ and proceed the chemical transformations expected on Fe surface to form fully deoxygenated BTX products[26,27]. The 1Pt@-10Fe@SiO$_2$ sample showed a guaiacol conversion and HDO selectivities nearly identical to those obtained on 10Fe@SiO$_2$ (Entry 5, Table 1), suggesting that the coexistence of Pt with Fe, but with spatial isolation achieved here, does not interfere with the catalytic properties of Fe in the hierarchically-porous SiO$_2$ framework. It has been reported that a close contact between Pt and Fe particles under the HDO reaction condition can improve the catalytic activity of Fe particles via hydrogen spillover from the Pt sites to the Fe ones[25,26], because the activity of Fe catalysts is generally limited by strongly bound O atoms on the Fe surfaces that hinder the activation of H$_2$ molecules[28]. This similarity of catalytic performance between 1Pt@-10Fe@SiO$_2$ and 10Fe@SiO$_2$ in guaiacol HDO (verified within 10–60`kPa H$_2$ partial pressures, Supplementary Fig. 8) thus provides compelling evidence for the unlikely hydrogen spillover between isolated Pt and Fe nanoparticles on the nonreducible SiO$_2$ support under the reaction conditions studied, which is consistent with previous reports[5,18].

Figure 4 shows that the formation rate of BTX ($r_{BTX}$) on 1Pt@-10Fe@SiO$_2$, a descriptor of the catalytic HDO activity, is sensitive to the H$_2$ partial pressure and the source of the H$_2$ feed (details of conversion and selectivity shown in Supplementary Table 1). As the H$_2$ partial pressure changed from 10 to 50 kPa, the $r_{BTX}$ value increased accordingly from 0.44 to 0.84 mol $mol_{Fe}^{-1}$ $h^{-1}$ consistent with the fact that H$_2$ is required for the oxygen removal from guaiacol. On the other hand, in lieu of feeding H$_2$ directly, H$_2$ is expected to be generated in situ via methanol steam reforming ($CH_3OH + H_2O \rightarrow CO_2 + 3H_2$) on Pt under the conditions studied[29,30], if CH$_3$OH and H$_2$O are co-fed into the reactor. We found that an equimolar mixture of CH$_3$OH and H$_2$O (4.8 kPa each) generated an effective H$_2$ partial pressure of 11 kPa (corresponding to 88% methanol conversion), which was measured from the effluent of the reactor (Supplementary Table 1). At this reaction condition, $r_{BTX}$ reached 1.28 mol $mol_{Fe}^{-1}$ $h^{-1}$, about three times the rate when H$_2$ was directly fed with a similar partial pressure (0.44 mol $mol_{Fe}^{-1}$ $h^{-1}$). This $r_{BTX}$ value achieved

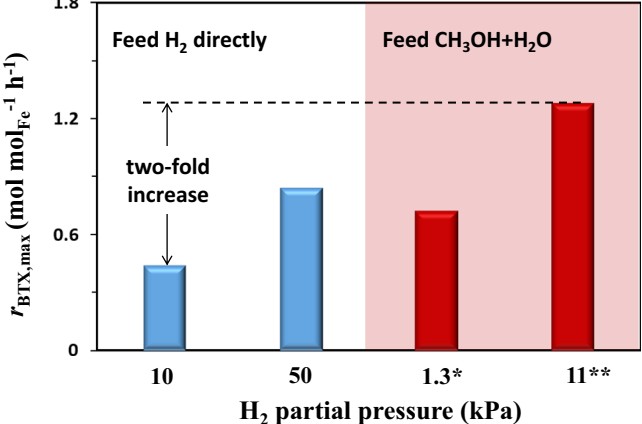

**Fig. 4 Effects of H$_2$ partial pressure and feed source on BTX formation rates.** Reaction condition: 450 °C, 0.5 kPa guaiacol, balanced by N$_2$, 0.25 g•s•mL$^{-1}$ space velocity, 1Pt@-10Fe@SiO$_2$ as catalyst. H$_2$ pressure was either directly produced via feeding gaseous H$_2$ or generated in situ via methanol stream reforming by cofeeding CH$_3$OH and H$_2$O instead of H$_2$. *Cofeeding 0.5 kPa CH$_3$OH and 0.5 kPa H$_2$O. **Cofeeding 4.8 kPa CH$_3$OH and 4.8 kPa H$_2$O.

with co-feeding CH$_3$OH and H$_2$O was even higher than that obtained with co-feeding 50 kPa H$_2$ (0.84 mol $mol_{Fe}^{-1}$ $h^{-1}$, Fig. 4). Such a promoting effect of the CH$_3$OH-H$_2$O mixture was further confirmed for guaiacol HDO at lower conversions (i.e., ~30%, Supplementary Fig. 9), in which PAC existed as the main products. Moreover, an activity enhancement of the guaiacol conversion was also observed for the 1Pt@SiO$_2$ catalyst (Supplementary Fig. 10), despite 1Pt@SiO$_2$ was much less active than 1Pt@-10Fe@SiO$_2$ and showed a high selectivity to PAC (Table 1). It is thus suggested that this promoting effect brought forth by the co-feeding of CH$_3$OH and H$_2$O appears to be general for the catalysts with spatially-separated H$_2$-activation and hydrodeoxygenation sites.

To better quantify the effects of H$_2$ generated in situ via methanol steam reforming, we further examined how low the CH$_3$OH and H$_2$O partial pressures could be to reach a $r_{BTX}$ value similar to that when directly feeding with 50 kPa H$_2$. As shown in Fig. 4, a comparable $r_{BTX}$ of 0.72 mol $mol_{Fe}^{-1}$ $h^{-1}$ was achieved when 0.5 kPa CH$_3$OH and 0.5 kPa H$_2$O were co-fed (vs 0.84 mol $mol_{Fe}^{-1}$ $h^{-1}$ for directly feeding with 50 kPa H$_2$), representing a much lower effective H$_2$ partial pressure of mere 1.3 kPa. It is worth noting that $r_{BTX}$ on 10Fe@SiO$_2$ was only about a quarter of that for 1Pt@-10Fe@SiO$_2$ when the same mixture of CH$_3$OH and H$_2$O was fed (Supplementary Fig. 11), which excludes possible roles of CH$_3$OH and H$_2$O directly involved in promoting guaiacol HDO on the accessible Fe particles (e.g., transfer hydrogenation

driven by methanol) and thus confirms that the Pt sites within the micropores of 1Pt@-10Fe@SiO$_2$ are responsible for the H$_2$ production from methanol stream reforming. Moreover, negligible formation of C$_1$-C$_2$ products in the presence of cofed CH$_3$OH and H$_2$O (Supplementary Table 1 of the supporting information) indicates that no Pt species migrated out from the small pores, otherwise significant C–C cleavage reactions would occur as observed for the Pt/C catalyst (71.9% selectivity, Table 1 of the manuscript). Here, we speculate that CH$_3$OH or H$_2$O may not only act as a hydrogen source during guaiacol HDO, but also transport active H species produced on the Pt sites to the Fe particles, accounting for the increased $r_{BTX}$.

**Hydrogen spillover enabled by molecular carriers over SiO$_2$.** Previous studies have reported that hydrogen spillover between two different metal sites on a nonreducible support is limited within the sub-nanometer scale and decays sharply as the distance increases[5,17], due to the unstable nature of atomic H species on the nonreducible support. To confirm our hypothesis on the promoted hydrogen spillover over Pt@-Fe@SiO$_2$ in the presence of the CH$_3$OH/H$_2$O mixture, we changed the distance between the Pt and Fe particles in the Pt@-10Fe@SiO$_2$ catalysts by varying the Pt loading and physically mixing the 1Pt@SiO$_2$ and 10Fe@SiO$_2$ catalysts (mass ratio 1:1; denoted as 1Pt@SiO$_2$ + 10 Fe@SiO$_2$) with different degrees of separation. As described above, the $x$Pt@-10Fe@SiO$_2$ catalysts ($x$ = 0.5, 1.0, 2.0 wt%) possessed Pt particles of similar sizes (~1.6 nm, Fig. 2 and Supplementary Fig. 7), which ensures that the increase of the Pt loading merely increased the areal density of the Pt particles and did not change their intrinsic activity. By assuming homogeneous distributions for both the Pt particles within the micropores and the Fe particles within the macropores, the average distances between the nearest Pt and Fe particles ($<d_{Pt-Fe}>$) in the $x$Pt@-10Fe@SiO$_2$ catalysts ($x$ = 0.5, 1.0, 2.0 wt%) are estimated to be 10, 5.7, and 3.4 nm, respectively. We further increased $<d_{Pt-Fe}>$ to the magnitude of 10$^3$ nm for the well-mixed 1Pt@SiO$_2$ + 10Fe@SiO$_2$ sample and 10$^7$ nm for the case with a dual-bed configuration (10Fe@SiO$_2$ as the downstream bed, Fig. 5). Guaiacol HDO on these encapsulated Pt-Fe catalysts with co-feeding CH$_3$OH and H$_2$O as the H$_2$ source showed that, even though the apparent H$_2$

partial pressures generated were similar (~1.3 kPa), $r_{BTX}$ increased exponentially from 0.17 to 0.92 mol mol$_{Fe}^{-1}$ h$^{-1}$ as $<d_{Pt-Fe}>$ decreased from 10$^7$ to 3.4 nm (Fig. 5), further confirming the existence of effective hydrogen spillover between spatially isolated Pt and Fe particles over SiO$_2$ brought forth by the CH$_3$OH or H$_2$O additive, especially in the length scale of several nm. It is also worth pointing out that the similar HDO rates obtained for the well-mixed and layered 1Pt@SiO$_2$ + 10Fe@SiO$_2$ catalysts (Fig. 5) exclude the possible role of gas-phase diffusion in the H transport process, since gaseous radicals can diffuse at a millimeter scale under similar conditions[31,32]. In other words, it is likely that the oxygenate-bound H radicals mainly transport over the SiO$_2$ surface (instead of diffusing in gas phase), benefiting from the stabilization of these radical by the oxide surface via H-bonding and van der Waals interactions.

We postulate that the transfer of active H atoms generated on the Pt sites to the Fe surface requires a molecular carrier, which should be capable of accepting the atomic H species to form a stable enough intermediate for long-distance transport and eventual H release. Intuitively, the most possible carrier candidates include not only CH$_3$OH and H$_2$O that were cofed to generate H$_2$ for the HDO reaction, but also formaldehyde (HCHO) that could form from dehydrogenation of CH$_3$OH on the Pt sites, although only a trace of HCHO was detected from the effluent because of the thermodynamic limit at the HDO reaction condition. Here, H-atom addition energy (HAE) was calculated via density functional theory (DFT) for these three candidates and used to assess their ability to carry a H atom. These HAE values were defined as the energy change for the addition of a H atom to a molecule (M) in gas phase,

$$HAE = E_{MH} - E_M - E_H \qquad (1)$$

in which $E_M$, $E_H$, and $E_{MH}$ are the energies for gaseous M, H, and the H-added MH product, respectively. According to DFT calculations (Fig. 6), the H-addition of H$_2$O is endothermic (HAE: 58 kJ mol$^{-1}$), even more unfavorable than that for H$_2$ (HAE: 16 kJ mol$^{-1}$). Although the H-addition of CH$_3$OH is exothermic (HAE: −125 kJ mol$^{-1}$, Fig. 6), CH$_3$OH is unstable by the attack of a H atom, leading to the formation of a methyl radical and a H$_2$O molecule. In contrast, HCHO is the most likely carrier for the hydrogen spillover on Pt@-Fe@SiO$_2$, because of the highly favorable nature of this H-addition process (HAE: −154 kJ mol$^{-1}$, Fig. 6) and the relatively high stability of the corresponding H-added product (i.e., CH$_2$OH·). The radical nature of such oxygenate transporting species during guaiacol HDO on 1Pt@-10Fe@SiO$_2$ was further confirmed by the observation of a

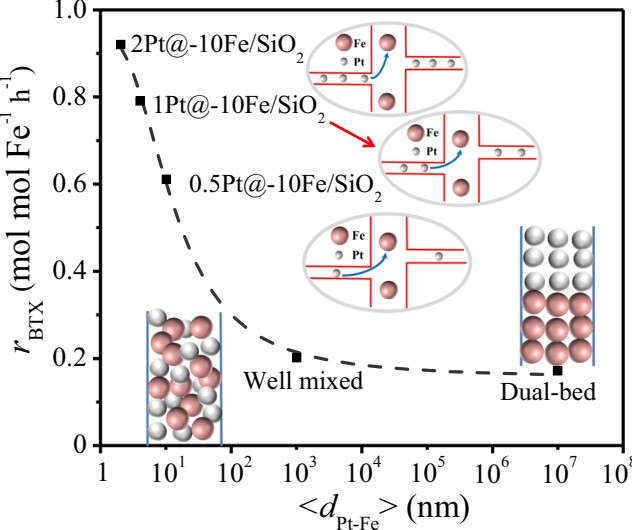

**Fig. 5 Effects of Pt-Fe distance on BTX formation rates.** $r_{BTX}$ is shown here as a function of $<d_{Pt-Fe}>$ for guaiacol HDO with H$_2$ generated in situ from methanol stream reforming (450 °C, 0.5 kPa guaiacol, 0.5 kPa methanol +0.5 kPa H$_2$O, balanced by N$_2$, 0.25 g•s•mL$^{-1}$ space velocity).

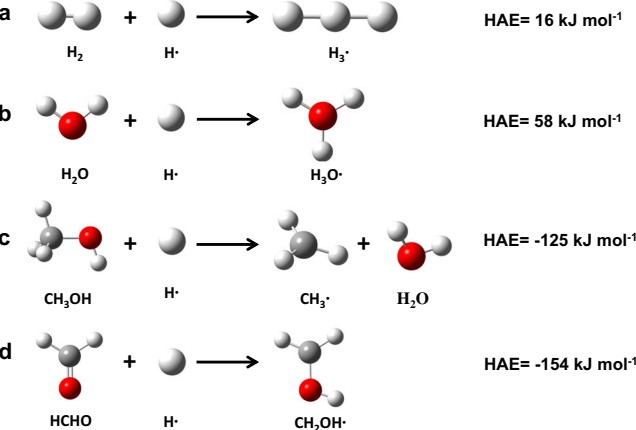

**Fig. 6 DFT-derived H-atom addition energies (HAE).** The calculations were carried out at the B3LYP/6-311 + G(d,p) level of theory.

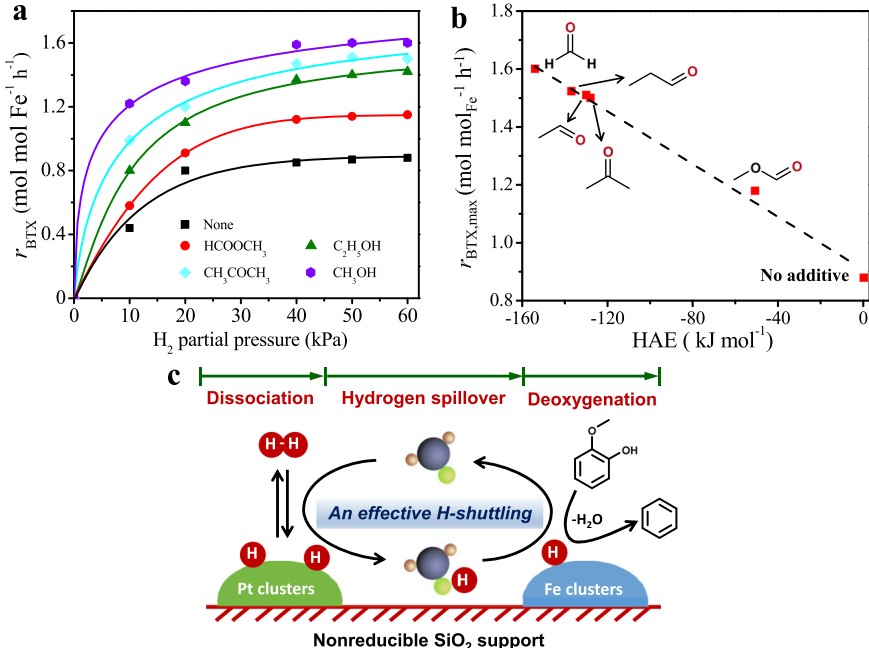

**Fig. 7 Promoting effects of gaseous oxygenate additives on BTX formation rates.** (**a**) a function of $H_2$ partial pressure with or without 0.5 kPa oxygenate additive. Reaction condition: 450 °C, 0.5 kPa guaiacol, balanced by $N_2$, methanol and ethanol used as the precursors of formaldehyde and acetaldehyde, respectively, 1Pt@-10Fe@SiO$_2$ as catalyst. Solid curves indicate trends. (**b**) correlation between the maximum $r_{BTX}$ ($r_{BTX,max}$) achieved on 1Pt@-10Fe@SiO$_2$ and DFT-derived HAE values for various oxygenate additives ($r_{BTX,max}$ taken from Fig. 7a; HAE taken from Supplementary Fig. 10). (**c**) An illustration of the proposed oxygenate-assisted hydrogen spillover mechanism over the inert SiO$_2$ support.

significant inhibiting effect of NO, a radical chain inhibitor, on the BTX formation rate no matter whether $H_2$ was directly fed or generated in situ from a mixture of $CH_3OH$ and $H_2O$ (Supplementary Figs. 12 and 13). It is also worth noting that the possible role of $H_2O$ in promoting the H transfer over $SiO_2$[9] was excluded based on the negligible effect of $H_2O$ addition on the HDO activity of 1Pt@-10Fe@SiO$_2$ at 10 kPa $H_2$ (Supplementary Fig. 14). On the other hand, a slight deactivation of the 1Pt@-10Fe@SiO$_2$ catalyst was observed with time-on-stream when cofeeding $CH_3OH$ and $H_2O$ (Supplementary Fig. 13), which is mainly due to the oxidation of metallic Fe by $H_2O$ or active site blocking by adsorbed oxygenate species[27,28]. This suggests that a sufficient $H_2$ partial pressure, which is either cofed or formed in situ, is requisite for preventing the Fe-based catalysts from deactivation during HDO.

The ability of carrying H atoms is hypothesized to be general for oxygenate molecules containing a carbonyl functional group (e.g., acetaldehyde, propanal, acetone, and methyl formate) based on their negative HAE values (DFT data shown in Supplementary Fig. 15). Therefore, we further examined the effects of these oxygenate additives on the rates of guaiacol HDO over 1Pt@-10Fe@SiO$_2$, and these experiments were conducted without cofeeding $H_2O$ in order to avoid the possible formation of surface OH groups on the SiO$_2$ support that may promote the hydrogen spillover[9]. It is also worth noting that the cofed $C_2$-$C_3$ oxygenates are stable and showed negligible conversions during the guaiacol HDO reaction, while the conversion of $CH_3OH$ is around 30%, mainly leading to the formations of $H_2$, HCHO and $CO_x$. As expected, all of them were able to promote the HDO rates (Fig. 7a), and the enhanced BTX formation rates approximately reached maximum values ($r_{BTX,max}$) as either the $H_2$ or the oxygenate additive partial pressure became sufficiently high (Fig. 7a and Supplementary Fig. 16, respectively), reflecting the intrinsic ability of these oxygenate additives in carrying H atoms under the HDO condition. As shown in Fig. 7b, $r_{BTX,max}$ increased with the oxygenate additives in the order of methyl

formate < acetone < acetaldehyde < propanal < formaldehyde, consistent with the trend of their HAE values. This excellent correlation strongly supports that these oxygenate additives containing a carbonyl functional group act as an effective H-carrier to promote guaiacol HDO on Fe surface (illustrated in Fig. 7c).

## Discussion

In summary, bimetallic Pt@-Fe@SiO$_2$ catalysts with Pt and Fe nanoparticles encapsulated in the respective micropores and macropores of the hierarchically structured SiO$_2$ support were prepared successfully via a dual template method and used here to examine the H transfer between these isolated metal sites during guaiacol hydrodeoxygenation. The distinct C–O/C–O hydrogenolysis selectivities of the Pt and Fe nanoparticles and the inaccessibility of the bulky guaiacol reactant to the micropores allowed us to experimentally confirm that H spillover over the SiO$_2$ surface is hard to occur at conditions relevant to catalysis, due to the nonreducible nature of SiO$_2$. More importantly, we found that the H migration from Pt to Fe sites can be dramatically enhanced when a small oxygenate additive (e.g., $C_1$-$C_3$ carbonyl compounds and esters) is present, resulting in much improved hydrodeoxygenation activity than the monometallic Fe and Pt catalysts. The stabilization of active H atoms conferred by the cofed oxygenate additive that underpins the achieved H spillover over nonreducible oxides also reflects from the decrease of activation barrier for the kinetically-relevant hydrogenating steps involved in guaiacol hydrodeoxygenation on Fe surface. It is noteworthy that these carbonyl H-carriers can be generated in situ from dehydrogenation or steam reforming of their alcohol precursors, which act concomitantly as a source of $H_2$ for the hydrodeoxygenation reaction. The above findings demonstrate a molecule-assisted strategy for establishing an efficient transfer of active chemical species between catalytic sites that are spatially isolated on an inert support, and may bring an alternative

approach to optimize or design novel systems of multifunctional catalysis.

## Methods

**Catalyst preparation**. The Pt@-Fe@SiO$_2$ catalysts with Pt and Fe particles deposited into the respective micropores and macropores of the hierarchically structured SiO$_2$ support were prepared via a dual-template step-impregnation method as described in our previous study[19]. In a typical preparation procedure, surfactant Tergitol 15-S-5 (used as the micropore template, Aldrich), concentrated HCl, and tetraethyl orthosilicate (TEOS, Sinopharm Chemical Reagent Co. 99.0%) were sequentially added into a mixed solvent of de-ionized water and ethanol (Sinopharm Chemical Reagent Co. 99.5%) with the molar ratio of TEOS: water: ethanol: HCl: Tergitol 15-S-5 kept at 1: 5-10: 5-10: 0.04: 0.05-0.1. After stirring for 24 h at ambient temperature, the resultant sol-gel solution was impregnated to cellulose extraction thimbles (used as the macropore template, 2.0 cm × 2.0 cm pieces, Whatman) and then dried at ambient temperature. These cellulose extraction thimbles were pre-treated at 350 °C (ramping rate 1 °C min$^{-1}$) for 1 h in flowing air before use. These impregnation and drying processes were repeated four times, followed by thermal treatment of the samples at 350 °C (ramping rate 2 °C min$^{-1}$) in flowing air for 4 h to remove the surfactant Tergitol 15-S-5 and form SiO$_2$/cellulose with micropores. Pt nanoparticles were deposited within these micropores of SiO$_2$/cellulose using the incipient wetness impregnation method and an aqueous solution of tetraamineplatinum nitrate (Pt(NH$_3$)$_4$(NO$_3$)$_2$, Aldrich, 99.995%) as the Pt precursor. The obtained solid was calcined in air at 550 °C (ramping rate 2 °C min$^{-1}$) for 5 h to decompose the Pt precursor and the cellulose template concurrently. Fe nanoparticles were then deposited within the created macropores via the incipient wetness impregnation (Fe(NO$_3$)$_3$ (Sinopharm Chemical Reagent Co., 98.5%) as the Fe precursor, followed by subsequent treatments in flowing N$_2$ (99.999%) at 350 °C (ramping rate 2 °C min$^{-1}$) for 5 h and in flowing 50 vol% H$_2$/N$_2$ at 450 °C (ramping rate 2 °C min$^{-1}$) for 2 h. These encapsulated bimetallic catalysts were denoted here as Pt@-Fe@SiO$_2$, in which the Pt content varied within 0–2.0 wt% and the Fe content was fixed at 0 or 10 wt%. For comparison, a nonporous carbon-supported Pt catalyst (3 wt% Pt/C) was also prepared by the incipient wetness impregnation method, and was calcined under N$_2$ at 350 °C for 5 h and then in flowing 50 vol% H$_2$/N$_2$ at 450 °C for 2 h before further use.

**Catalyst characterization**. Powder X-ray diffraction (XRD) measurements were performed by scanning within 2$\theta$ ranges of 10°–90° (10° min$^{-1}$) and 0.5°–5.0° (5° min$^{-1}$) on a Rigaku Ultima IV apparatus using Cu K$\alpha$ radiation ($\lambda$ = 0.15418 nm) operated at a voltage of 40 kV and a current of 30 mA. The specific surface area and average pore size of each sample were determined form N$_2$ adsorption-desorption isotherms obtained on a Micromeritics ASAP 2020 physisorption analyzer at liquid nitrogen temperature (−196 °C), in which each sample was degassed at 150 °C for 4 h before measurements. Images of transmission electron microscopy (TEM) were collected with a Tecnai G2 F20 field emission microscope operating at 200 kV, and the size distributions of the Pt and Fe particles were obtained by counting ca. 200 particles for each in the images. H$_2$-temperature programmed reduction (H$_2$-TPR) experiments were carried out in a homemade fixed-bed reactor. 100 mg of sample was in situ pre-treated in flowing Ar (30 mL min$^{-1}$) at 200 °C for 1 h before test. After cooling down to 30 °C, the sample was heated to 800 °C (ramping rate 10 °C min$^{-1}$) in a 5 vol% H$_2$/Ar flow (30 mL min$^{-1}$), in which the H$_2$ uptake was measured and quantified using a thermal conductivity detector (TCD).

**Catalytic tests**. Guaiacol hydrodeoxygenation was conducted under atmospheric pressure in a vertical fixed-bed quartz tube reactor with an internal diameter of 8 mm and a length of 750 mm. 200 mg of catalyst diluted with 800 mg of quartz particles (0.18–0.25 mm) was placed between two layers of quartz wool in the center of the reactor. The temperature of the catalyst was monitored using a K-type thermocouple placed in the middle of the catalyst bed. Prior to catalytic measurements, the catalyst was treated in flowing 50 vol% H$_2$/N$_2$ (50 ml min$^{-1}$) at 450 °C (ramping rate 5 °C min$^{-1}$) for 2 h. Guaiacol and oxygenate additives with set partial pressures were introduced separately using syringe pumps and transferred into the reactor by a N$_2$ flow. All transfer lines were heated to around 210 °C to avoid any condensation of liquid reactants or products. The components of the effluent from the reactor were analyzed quantitatively with an on-line gas chromatograph with a DB-WAXETR column (50 m × 0.320 mm) connected to a flame ionization detector (FID) for separating oxygenates and hydrocarbons and a TDX-01 column connected to a thermal conductivity detector (TCD) for separating small gassous molecules (e.g., CH$_4$, CO and CO$_2$ and H$_2$). N$_2$ was used as an internal standard, and response factors for guaiacol and the hydrodeoxygenation products were determined using standard chemicals with known concentrations. The mass balance for each experiment, unless otherwise noted, was at least 95%. Guaiacol conversions and product selectivities were reported on a carbon basis as described elsewhere[25].

**Theoretical calculation**. The values of hydrogen affinity (Eq. (1)) for various oxygenates, H$_2$O, and H$_2$ were calculated via the Gaussian software package[33] at the hybrid B3LYP functional level of theory[34,35]. The standard 6−311 + G(d,p) basis set[36,37] was used for all atoms, and the Grimme's D3BJ dispersion correction[38] was taken into account in the electronic energy calculations. Geometry optimizations were performed using the Berny geometry algorithm[39] with convergence criteria of $1.0 \times 10^{-8}$ Ha for energy and $1.5 \times 10^{-5}$ Ha Bohr$^{-1}$ for the maximum residual forces on each atom.

## Data availability
The authors declare that all the relevant data within this paper and its Supplementary Information file are available from the corresponding authors upon a reasonable request.

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

## Acknowledgements

This work was supported by the National Natural Science Foundation of China (No. 21922201 (Shu. W.), 21872113 (Shu. W.), 22121001 (Shu. W.), 91945301 (Shu. W.), and 91545114 (Y. W.)), the Postdoctoral Science Foundation of China (No. 2016M600501 (M. T.)), and the Fundamental Research Funds for the Central Universities (No. 20720190036 (Shu. W.)).

## Author contributions

Y.W., Shu.W., and M.T. conceived the idea for the project. M.T., Ya.Y., Yi.Y. and J.C. conducted the catalyst synthesis, structural characterization, and catalytic experiments. Shu.W. conducted the theoretical calculations. M.T. drafted the manuscript under the guidance of Y.W and Shu.W. Z.Z., G.F., J.L., Sha.W. and the authors mentioned above all discussed and commented on the manuscript.

## Competing interests

The authors declare no competing interests.
