## [Peer Review File · Nature Communications]

Title: Hydrogen spillover assisted by oxygenate molecules over nonreducible oxidesReviewers' comments:

Reviewer #1 (Remarks to the Author):

In the present work, the authors synthesized hierarchically micro-/macroporous SiO₂, in which Pt is mainly encapsulated within micropores and Fe is located within macropores, respectively. The authors claimed that various oxygenates containing carbonyl groups, such as formaldehyde and acetaldehyde, can enhance the hydrodeoxygenation activity of guaiacol by enhancing catalytic communication between remote Pt and Fe. I am rather confused, because I understand that hydrogen spillover is the phenomenon where the activated hydrogen migrates over the "solid" support surface. In the present work, the "gaseous" unsaturated oxygenate molecules shuttle between Pt and Fe particles and repeat redox (hydrogenation-dehydrogenation) cycles to carry hydrogen atoms. This is conceptually more close to transfer hydrogenation.

Although the authors claimed that this study provides important insight for understanding hydrogen spillover over the irreducible oxide surface (the topic of controversy in heterogeneous catalysis), I feel that this study is just one of many studies about guaiacol hydrodeoxygenation. Catalytic studies are too primitive for publication in high-profile journals. What are the fate of oxygenate additives during the reaction? Are they completely stable during the reaction or consumed? There is no careful kinetic and spectroscopic investigation to understand surface reactions. The main conclusion was mainly based on the DFT calculation of H-atom addition energy.

The present study is partly interesting, but I feel that it is more suitable for publication in specialized journals, such as ChemSusChem or ACS Sustainable Chemistry and Engineering.

Reviewer #2 (Remarks to the Author):

Report on Hydrogen spillover assisted by oxygenate molecules over nonreducible oxides by M. Tan et al., NCOMMS-21-30739

The authors show that a Pt@-Fe@SiO₂ catalyst with Pt mainly in micropores and Fe in macropores has a similar activity and selectivity in the conversion of guaiacol as a Fe@SiO₂ catalyst. However, the presence of oxygen-containing increased the catalytic activity. This was ascribed to a spillover effect of hydrogenated organic molecule by transport through the gas phase. DFT calculations of the stability of the hydrogenated organics backed this up.

The presented results are new, very interesting and should be published. The authors should, however, add some discussion of two alternative explanations besides their hydrogen spillover explanation. Some twenty years ago Sachtler and coworkers showed that Pt particles inside the micropores of zeolites move to Fe particles at the outer surface when an oxidation treatment is applied, see J. Catal. 188 (1999) 365, J. Catal. 191 (2000) 364, and Topics Catal. 10 (2000) 49. The reason is that PtO₂ is volatile. In this context, wouldn't it be possible that oxygen-containing molecules oxidize the Pt to PtO₂, which then comes close to the Fe?

A second explanation which should be discussed is hydrogen transfer. Wouldn't it be possible that organic molecules which contain oxygen can react with the molecules that must undergo hydrodeoxygenation by H-transfer on the Fe particles in the macropores? The reduced organic molecule then diffuses into the micropores where it is reduced back to the organic molecule on a Pt particle. For instance, methanol reacts with guaiacol or a derivative on Fe and the resulting formaldehyde is hydrogenated back to methanol on Pt in the micropores. Like in the explanation of the authors, a hydrogen shuttle takes place, but instead of a hydrogenated organic molecule (e.g. CH₃OH₂) the organic molecule itself (e.g. CH₃OH) transports the H atom. In that case hydrogen spillover does not have to be invoked.

Minor remarks:

The names of the molecules at the bottom of fig. 1 are almost illegible, they should be enlarged. Line 323 is not clear. Is this pretreatment at 350 °C for 1 h applied to the cellulose thimbles, or is it applied to the thimbles after they were impregnated with the sol-gel solution? From their own reference 18 I got the impression it is the first.

I believe that it is more prudent to write "of their support, known as spillover, has been claimed to play a major role" on line 16.

Some typos: line 19 challenged, line 59 steam reforming, line 61 remained, line 156 PAC instead of POA, line 339 nonporous carbon-supported.

Reviewer #3 (Remarks to the Author):

In this paper the HDO of guaiacol was performed using hierarchically-porous-SiO₂-supported bimetallic Pt-Fe catalysts with Pt nanoparticles exclusively deposited into the micropores. The authors claim that activated hydrogen species generated on the Pt sites within the micropores can be readily transported by oxygenate molecules to Fe sites located in macropores, leading to significantly accelerated hydrodeoxygenation rates on the latter sites. According to the authors this finding provides a molecule-assisted approach to the rational design and optimization of multifunctional heterogeneous catalysts, reminiscent of the role of molecular coenzymes in bio-catalysis.

These statements are based on the comparison of the HDO activity of SiO₂, 1Pt@SiO₂, 10Fe@SiO₂ and 1Pt@-10Fe@SiO₂ catalysts. The existence of spatially separated Pt and Fe nanoparticles was demonstrated by H₂/TPR experiments. The active H species were supplied by either gaseous H₂ or steam-reforming of co-fed CH₃OH+H₂O. The use of steam-reforming mixture gave highly enhanced HDO activity and this led the authors to postulate the existence of active H atoms transported with the aid of gaseous organic molecules containing a carbonyl group. They excluded any intervention of hydrogen spillover, believed to be forbidden for nonreducible oxides.

The proposal of oxygenates species as transporting agents of H atoms in a hydroconversion reaction is original and the reasoning based on substantial results is correct. However, close inspection of the paper shows that the results and interpretation need to be debated.

1. From the fundamental point of view, the hydrogen spillover effect was recently reviewed and shown

to be effective for nonreducible oxides [Bettahar, Catal. Rev., 2021]. In addition, several examples were reported by the authors. In references 9 and 10 it was shown that Pt nanoparticles encapsulated in acidic zeolites hydrogenated benzene although this molecule was inaccessible to the metal phase. Similar effect was presented in reference 11 for the hydrogenation of acetylene to ethylene over Pd nanoparticles encapsulated in sodalite zeolites. In reference 16 it was concluded that CO₂ methanation kinetics over PtCo/SiO₂ catalysts (where the Pt and Co nanoparticles were separated) exhibited diffusion limitation of the hydrogenating H species spilling from Pt to Co sites on the support. In a non reported paper, Nishiyama et al. (J. Catal. 2012) performed the selective hydrogenation of crotonaldehyde to the unsaturated crotyl alcohol on a Sn-modified SiO₂-coated Pt catalyst. TEM and EXAFS experiments showed that Sn species were separated from the core Pt nanoparticles. The results obtained indicated that the C=O group was hydrogenated on Sn sites by H spillover atoms formed on Pt sites. The PtSn system is very similar to the PtFe system used in the paper presently submitted. Finally, the rebuttal of hydrogen spillover for nonreducible oxides reported in references 5 and 17 was demonstrated to be based on non realistic experimental conditions (Bettahar, Catal. rev., 2021).

An important finding in the review cited is that the structure of hydrogen spillover consists in H/OH pairs formed by dehydroxylation processes during the metal phase reduction (see Scheme below). As a result their reactivity and concentration increases with that of the support surface hydroxyls and also with temperature and time of reduction. Therefore the more acidic/hydroxylated the support the more intense is the spillover effect.

2. The HDO activity of 1Pt@SiO₂ and SiO₂ were similar, indicating the absence of hydrogen spillover. We agree with the authors that the postulated oxygenate species transporting hydrogen atoms may be considered. Nevertheless, close inspection of the results, shows that 1Pt@SiO₂, is a little more active (15.8%) than SiO₂ (11.2%), attributable to a small hydrogen spillover effect contribution. It is worth noting that compared to alumina [Teichner, Appl. catal., 1990] or zeolites [Benseraj, Appl. Catal., 2002] silica is less prone to hydrogen spillover effect. This could be the case here.

3. When H₂ was supplied by the steam-reforming reaction (instead of gaseous H₂) and using 1Pt@-10Fe@SiO₂ as a catalyst, enhanced catalytic activity could be due to the presence of the water molecules added. This was shown in the gas phase hydrogenation of benzaldehyde with gold catalysts supported on alumina and in the presence of water vapor [Perret, Catal. Comm., 2011]. Water dissociation produces hydroxyls which serve as bridges for the hydrogen spillover species transport on the support surface. This phenomenon has long been known [Benson, J. Catal., 1966; Lenz, J. Catal., 1988]. This may apply for the HDO of guaiacol. The H active species produced in the micropores on Pt could directly migrate to the macropores by the spillover effect with the help of H₂O. There is no need of oxygenate molecules to their transport.

If it is such the higher reaction rate (based on BTX production) of 1Pt@-10Fe@SiO₂ as compared to that of 10Fe@SiO₂ catalyst can be explained also by the enhanced spillover effect due to the water molecules. Similarly the additional amounts of activated hydrogen do vary with the composition of the co-fed mixture (Figure 4) and the nature of the oxygenate molecule (Figure 7). In the same way the activity of 1Pt@-10Fe@SiO₂ decrease with the increase of the distance <dPt-Fe> between Pt and iron (Figure 5). This is an expected process for hydrogen spillover whatever the nature of its source.

4. From the above considerations it would be of interest to perform the HDO reaction over 1Pt@SiO₂ catalyst using CH₃OH+H₂O co-feeding mixture and compare to reaction using H₂ co-feeding.

5. If the oxygenate transporting species exist then they could be evidenced by radical chain inhibitors, e. g. NO.

Response to Reviewers

We appreciate greatly all the three reviewers' time and effort for improving this manuscript and have provided the point-by-point response to their comments below:

Response to Reviewer 1:

Reviewer 1: *In the present work, the authors synthesized hierarchically micro-/macroporous SiO₂, in which Pt is mainly encapsulated within micropores and Fe is located within macropores, respectively. The authors claimed that various oxygenates containing carbonyl groups, such as formaldehyde and acetaldehyde, can enhance the hydrodeoxygenation activity of guaiacol by enhancing catalytic communication between remote Pt and Fe. I am rather confused, because I understand that hydrogen spillover is the phenomenon where the activated hydrogen migrates over the “solid” support surface. In the present work, the “gaseous” unsaturated oxygenate molecules shuttle between Pt and Fe particles and repeat redox (hydrogenation-dehydrogenation) cycles to carry hydrogen atoms. This is conceptually more close to transfer hydrogenation.*

Response: We are afraid that Reviewer 1 might have misunderstood the oxygenate-assisted hydrogen transport process we are trying to report here. Transfer hydrogenation refers to the addition of hydrogen to a molecule from a source other than gaseous H₂ (such as formic acid or alcohols). This is clearly not the case reported here, because we do see a significantly higher hydrodeoxygenation rate on Pt-Fe than Fe with methanol as shown in Fig. S10 of the Supplementary Information (attached below as Fig. R1). If transfer hydrogenation does exist, we would expect to see similar initial hydrodeoxygenation rates since methanol can access the Fe surface in both cases. Another evidence is that the hydrodeoxygenation activity can be enhanced as well when methyl formate or acetone is cofed instead of the mixture of methanol and H₂O (Fig. R2, adopted from Fig. 7A of the manuscript). The conversions of methyl formate and acetone are negligible under the reaction condition of our study, and the H atoms in these two additives are not available for transfer hydrogenation. Therefore, the observed promoting effects of methyl formate and acetone in Fig. R2 also allow us to rule out the possible contribution of transfer hydrogenation. To avoid the potential confusion, we have revised the relevant discussion in the manuscript (highlighted in yellow, Page 9):

“It is worth noting that r_{BTX} on 10Fe@SiO₂ was only about a quarter of that for 1Pt@-10Fe@SiO₂ when the same mixture of CH₃OH and H₂O was fed (Fig. S10), which excludes possible roles of CH₃OH

and H₂O directly involved in promoting guaiacol HDO on the accessible Fe particles (e.g. transfer hydrogenation driven by methanol) and thus confirms that the Pt sites within the micropores of 1Pt@-10Fe@SiO₂ are responsible for the H₂ production from methanol stream reforming...”

Fig. R1. Comparison of the BTX formation rate (r_{BTX}) in guaiacol hydrodeoxygenation between 10Fe@/SiO₂ and 1Pt@-10Fe@SiO₂ when CH₃OH and H₂O were cofed instead of H₂. Reaction condition: 450 °C, 0.5 kPa guaiacol, 0.5 kPa CH₃OH, 0.5 kPa H₂O, balanced by N₂, 0.25 g.s.mL⁻¹ space velocity. Adopted from Fig. S10 of the supporting information.

Fig. R2. The formation rate of BTX (r_{BTX}) as a function of H₂ partial pressure with or without 0.5 kPa oxygenate additive for guaiacol hydrodeoxygenation on 1Pt@-10Fe@SiO₂. Reaction condition: 450 °C, 0.5 kPa guaiacol, balanced by N₂, methanol and ethanol used as the precursors of formaldehyde and acetaldehyde, respectively. Solid curves indicate trends. Adopted from Fig. 7A of the manuscript.

In addition, even though the cofed oxygenate molecules mainly exist in the gaseous state under the reaction condition of hydrodeoxygenation, it does not appear that the hydrogen transport between the Pt and Fe sites assisted by these oxygenates occurs via gas-phase diffusion. It has been well reported that gaseous radicals diffuse at a millimeter scale under similar conditions (*J. Phys. Chem. C*, 2021, 125, 5623;

Org. Process Res. Dev., 2018, 22, 1644). Therefore, the mixing degree of 1Pt@SiO₂ and 10Fe@SiO₂ is expected to affect the hydrodeoxygenation activity of their physical mixtures if the gas-phase diffusion pathway prevails for the hydrogen transport between the Pt and Fe sites. However, very similar hydrodeoxygenation rates were obtained when the Pt@SiO₂ and 10Fe@SiO₂ particles were well-mixed or layered in the catalyst bed within the millimeter scale (Fig. 5 of the manuscript; attached here as Fig. R3). Furthermore, the hydrodeoxygenation rates increased significantly only when Pt and Fe sites were deposited on the same SiO₂ support, e.g., Pt@-10Fe@SiO₂ catalysts (Fig. R3). These results suggest that the oxygenate-bound H radicals prefer to transport over the SiO₂ surface (instead of diffusing in gas phase), which is probably due to the stabilization of these radical by the oxide surface via H-bonding and van der Waals interactions. We have added a statement about this point in the revised manuscript as follows (Page 10):

“It is also worth pointing out that the similar HDO rates obtained for the well-mixed and layered 1Pt@SiO₂+10Fe@SiO₂ catalysts (Fig. 5) exclude the possible role of gas-phase diffusion in the H transport process, since gaseous radicals can diffuse at a millimeter scale under similar conditions^{31,32}. In other words, it is likely that the oxygenate-bound H radicals mainly transport over the SiO₂ surface (instead of diffusing in gas phase), benefiting from the stabilization of these radical by the oxide surface via H-bonding and van der Waals interactions.”

Fig. R3. Effects of Pt-Fe distance and oxygenate additive on hydrodeoxygenation activity. r_{BTX} is shown here as a function of $\langle d_{Pt-Fe} \rangle$ for guaiacol HDO with H₂ generated *in situ* from methanol stream reforming (450 °C, 0.5 kPa guaiacol, 0.5 kPa methanol + 0.5 kPa H₂O, balanced by N₂, 0.25 g·s·mL⁻¹ space velocity). Adopted from Fig. 5 of the manuscript.

Reviewer 1: *Although the authors claimed that this study provides important insight for understanding hydrogen spillover over the irreducible oxide surface (the topic of controversy in heterogeneous catalysis), I feel that this study is just one of many studies about guaiacol hydrodeoxygenation. Catalytic studies are too primitive for publication in high-profile journals. What are the fate of oxygenate additives during the reaction? Are they completely stable during the reaction or consumed? There is no careful kinetic and spectroscopic investigation to understand surface reactions. The main conclusion was mainly based on the DFT calculation of H-atom addition energy.*

The present study is partly interesting, but I feel that it is more suitable for publication in specialized journals, such as ChemSusChem or ACS Sustainable Chemistry and Engineering.

Response: As addressed in the response above, our study focuses on reporting an enabling strategy to achieve effective hydrogen spillover on nonreducible metal oxides, instead of guaiacol hydrodeoxygenation, which are praised by both of Reviewer 2 and Reviewer 3 in recognizing the novelty of our strategy. In particular, Reviewer 2 remarked that “*the presented results are new, very interesting and should be published*”, and Reviewer 3 remarked that “*the proposal of oxygenates species as transporting agents of H atoms in a hydroconversion reaction is original and the reasoning based on substantial results is correct*”. We would also like to emphasize that we have attempted to provide sufficient experimental data to confirm the existence of oxygenate-assisted hydrogen spillover on the Pt-Fe/SiO₂ catalysts. The DFT calculations were employed here to validate a plausible mechanism for the observed hydrogen transport process. Detailed kinetic and spectroscopic studies have been previously reported for the same catalytic systems (e.g., *ACS Catal.*, 2017, 7, 3639 and *Catal. Commun.*, 2017, 100, 43), and there is no evidence suggesting that the pathway of guaiacol hydrodeoxygenation on the Fe surface is altered. Similar trends for the dependence of hydrodeoxygenation rates on H₂ pressure were observed with and without cofed oxygenates (Fig. R2). Regarding the fate of the oxygenate additives during these hydrodeoxygenation reactions, we have shown that, under the conditions studied, the C₂-C₃ oxygenates are stable and show negligible conversions, while the conversion of CH₃OH is around 30%, mainly leading to the formations of H₂, HCHO and CO_x. This information has been included in the revised manuscript as follows (Page 13):

“...It is also worth noting that the cofed C₂-C₃ oxygenates are stable and show negligible conversions during the guaiacol HDO reaction, while the conversion of CH₃OH is around 30%, mainly leading to the formations of H₂, HCHO and CO_x.”

Response to Reviewer 2:

Reviewer 2: *The authors show that a Pt@-Fe@SiO₂ catalyst with Pt mainly in micropores and Fe in macropores has a similar activity and selectivity in the conversion of guaiacol as a Fe@SiO₂ catalyst. However, the presence of oxygen-containing increased the catalytic activity. This was ascribed to a spillover effect of hydrogenated organic molecule by transport through the gas phase. DFT calculations of the stability of the hydrogenated organics backed this up.*

The presented results are new, very interesting and should be published. The authors should, however, add some discussion of two alternative explanations besides their hydrogen spillover explanation. Some twenty years ago Sachtler and coworkers showed that Pt particles inside the micropores of zeolites move to Fe particles at the outer surface when an oxidation treatment is applied, see J. Catal. 188 (1999) 365, J. Catal. 191 (2000) 364, and Topics Catal. 10 (2000) 49. The reason is that PtO₂ is volatile. In this context, wouldn't it be possible that oxygen-containing molecules oxidize the Pt to PtO₂, which then comes close to the Fe?

Response: Thank you for the positive comment and mentioning these references. We are very well aware of the oxidation and redispersion of Pt under oxidative conditions including our recent publications (e.g. *Science*, 2016, 353, 150). But we have unambiguous evidence to exclude the migration of PtO₂ during the hydrodeoxygenation reaction. First, Pt@-Fe@SiO₂ showed negligible selectivities to C₁-C₂ products (e.g. methane, CO, CO₂, and ethane) with or without the oxygenate additives (Table S1 of the supporting information). If Pt migrates out from the small pores, guaiacol will access the Pt sites and result in high selectivities to the C₁-C₂ molecules because of the strong C-C cleavage ability of Pt as observed for the Pt/C catalyst (71.9% selectivity, Table 1 of the manuscript). Second, if Pt does migrate to the Fe surface, we wouldn't be able to see the effects of Pt-Fe distance on the hydrodeoxygenation reactivity shown in Fig. 5 of the manuscript. It is also worth noting that the hydrodeoxygenation reaction occurred at 450 °C and in the presence of H₂, which created a strongly reductive atmosphere to keep Pt in the metallic state. To address this potential concern, we have added a relevant discussion in the manuscript to strengthen our conclusion as follows (Page 9):

“Moreover, negligible formation of C₁-C₂ products in the presence of cofed CH₃OH and H₂O (Table S1 of the supporting information) indicates that no Pt species migrated out from the small pores, otherwise significant C-C cleavage reactions would occur as observed for the Pt/C catalyst (71.9% selectivity, Table 1 of the manuscript).”

Reviewer 2: A second explanation which should be discussed is hydrogen transfer. Wouldn't it be possible that organic molecules which contain oxygen can react with the molecules that must undergo hydrodeoxygenation by H-transfer on the Fe particles in the macropores? The reduced organic molecule then diffuses into the micropores where it is reduced back to the organic molecule on a Pt particle. For instance, methanol reacts with guaiacol or a derivative on Fe and the resulting formaldehyde is hydrogenated back to methanol on Pt in the micropores. Like in the explanation of the authors, a hydrogen shuttle takes place, but instead of a hydrogenated organic molecule (e.g. CH_3OH_2) the organic molecule itself (e.g. CH_3OH) transports the H atom. In that case hydrogen spillover does not have to be invoked.

Response: As also pointed by Reviewer 1 above, the hydrogenation process driven by the methanol-formaldehyde redox cycles can be regarded as transfer hydrogenation. However, we find that the hydrodeoxygenation activity on the Pt-Fe catalyst can also be enhanced by cofeeding methyl formate or acetone instead of methanol (Fig. R4, adopted from Fig. 7A of the manuscript), in which these two additives show negligible conversions and the H atoms in them are not available for transfer hydrogenation under the condition studied. Therefore, the observed promoting effects of methyl formate and acetone rule out the possible role of transfer hydrogenation. We would also like to mention that, if transfer hydrogenation does exist, we would expect to see similar initial hydrodeoxygenation rates for the Pt-Fe and Fe catalysts with methanol since methanol can access the Fe surface in both cases. The apparently higher hydrodeoxygenation rate on Pt-Fe than Fe with methanol (Fig. R5, adopted from Fig. S10 of the Supplementary Information) thus excludes the contribution of transfer hydrogenation as well. In addition, as we responded to the comment by Reviewer 1 above, the hydrodeoxygenation rates increased significantly only when Pt and Fe sites were deposited on the same SiO_2 support, e.g., Pt@-10Fe@ SiO_2 catalysts. This further suggests that the oxygenate-bound H radicals prefer to transport over the SiO_2 surface as opposed to methanol/formaldehyde transport in gas phase speculated by Reviewer 2. We have revised the relevant discussion in the manuscript (highlighted in yellow, Page 9):

“It is worth noting that r_{BTX} on 10Fe@ SiO_2 was only about a quarter of that for 1Pt@-10Fe@ SiO_2 when the same mixture of CH_3OH and H_2O was fed (Fig. S10), which excludes possible roles of CH_3OH and H_2O directly involved in promoting guaiacol HDO on the accessible Fe particles (e.g. transfer hydrogenation driven by methanol) and thus confirms that the Pt sites within the micropores of 1Pt@-10Fe@ SiO_2 are responsible for the H_2 production from methanol stream reforming...”

Fig. R4. The formation rate of BTX (r_{BTX}) as a function of H₂ partial pressure with or without 0.5 kPa oxygenate additive for guaiacol hydrodeoxygenation on 1Pt@-10Fe@SiO₂. Reaction condition: 450 °C, 0.5 kPa guaiacol, balanced by N₂, methanol and ethanol used as the precursors of formaldehyde and acetaldehyde, respectively. Solid curves indicate trends. Adopted from Fig. 7A of the manuscript.

Fig. R5. Comparison of the BTX formation rate (r_{BTX}) in guaiacol hydrodeoxygenation between 10Fe@/SiO₂ and 1Pt@-10Fe@SiO₂ when CH₃OH and H₂O were cofed instead of H₂. Reaction condition: 450 °C, 0.5 kPa guaiacol, 0.5 kPa CH₃OH, 0.5 kPa H₂O, balanced by N₂, 0.25 g.s.mL⁻¹ space velocity. Adopted from Fig. S10 of the supporting information.

Reviewer 2: Minor remarks:

The names of the molecules at the bottom of fig. 1 are almost illegible, they should be enlarged.

Response: Thank you for your suggestion. Fig. 1 has been revised accordingly (Fig. R6).

Fig. R6. Hydrogen spillover enhanced by oxygenate additives during catalysis. An oxygenate molecule (e.g. formaldehyde) acts as an H-carrier to promote hydrogen spillover for guaiacol hydrodeoxygenation on hierarchically porous Pt@-Fe@SiO₂ catalysts.

Reviewer 2: Line 323 is not clear. Is this pretreatment at 350 °C for 1 h applied to the cellulose thimbles, or is it applied to the thimbles after they were impregnated with the sol-gel solution? From their own reference 18 I got the impression it is the first.

Response: Sorry for the ambiguous expression. The sentence has been revised as:

“After stirring for 24 h at ambient temperature, the resultant sol-gel solution was impregnated to cellulose extraction thimbles (used as the macropore template, 2.0 cm × 2.0 cm pieces, Whatman) and then dried at ambient temperature. These cellulose extraction thimbles were pre-treated at 350 °C (ramping rate 1 °C min⁻¹) for 1 h in flowing air before use.”

Reviewer 2: I believe that it is more prudent to write “of their support, known as spillover, has been claimed to play a major role” on line 16.

Response: Thank you for this suggestion. The sentence has revised accordingly.

Reviewer 2: Some typos: line 19 challenged, line 59 steam reforming, line 61 remained, line 156 PAC instead of POA, line 339 nonporous carbon-supported.

Response: Thank you for the careful proofreading. These typos have been corrected in the revised manuscript.

Response to Reviewer 3:

Reviewer 3: *In this paper the HDO of guaiacol was performed using hierarchically-porous-SiO₂-supported bimetallic Pt-Fe catalysts with Pt nanoparticles exclusively deposited into the micropores. The authors claim that activated hydrogen species generated on the Pt sites within the micropores can be readily transported by oxygenate molecules to Fe sites located in macropores, leading to significantly accelerated hydrodeoxygenation rates on the latter sites. According to the authors this finding provides a molecule-assisted approach to the rational design and optimization of multifunctional heterogeneous catalysts, reminiscent of the role of molecular coenzymes in bio-catalysis.*

These statements are based on the comparison of the HDO activity of SiO₂, 1Pt@SiO₂, 10Fe@SiO₂ and 1Pt@-10Fe@SiO₂ catalysts. The existence of spatially separated Pt and Fe nanoparticles was demonstrated by H₂/TPR experiments. The active H species were supplied by either gaseous H₂ or steam-reforming of co-fed CH₃OH+H₂O. The use of steam-reforming mixture gave highly enhanced HDO activity and this led the authors to postulate the existence of active H atoms transported with the aid of gaseous organic molecules containing a carbonyl group. They excluded any intervention of hydrogen spillover, believed to be forbidden for nonreducible oxides.

The proposal of oxygenates species as transporting agents of H atoms in a hydroconversion reaction is original and the reasoning based on substantial results is correct. However, close inspection of the paper shows that the results and interpretation need to be debated.

1. From the fundamental point of view, the hydrogen spillover effect was recently reviewed and shown to be effective for nonreducible oxides [Bettahar, Catal. Rev., 2021]. In addition, several examples were reported by the authors. In references 9 and 10 it was shown that Pt nanoparticles encapsulated in acidic zeolites hydrogenated benzene although this molecule was inaccessible to the metal phase. Similar effect was presented in reference 11 for the hydrogenation of acetylene to ethylene over Pd nanoparticles encapsulated in sodalite zeolites. In reference 16 it was concluded that CO₂ methanation kinetics over PtCo/SiO₂ catalysts (where the Pt and Co nanoparticles were separated) exhibited diffusion limitation of the hydrogenating H species spilling from Pt to Co sites on the support. In a non reported paper, Nishiyama et al. (J. Catal. 2012) performed the selective hydrogenation of crotonaldehyde to the unsaturated crotyl alcohol on a Sn-modified SiO₂-coated Pt catalyst. TEM and EXAFS experiments showed that Sn species were separated from the core Pt nanoparticles. The results obtained indicated that the C=O group was hydrogenated on Sn sites by H spillover atoms formed on Pt sites. The PtSn system is very similar to the PtFe system used in the paper presently submitted. Finally, the rebuttal of hydrogen spillover for nonreducible oxides reported in references 5 and 17 was demonstrated to be based on non realistic experimental conditions (Bettahar, Catal. rev., 2021).

An important finding in the review cited is that the structure of hydrogen spillover consists in H/OH pairs formed by dehydroxylation processes during the metal phase reduction (see Scheme below). As a result, their reactivity and concentration increase with that of the support surface hydroxyls and also with temperature and time of reduction. Therefore, the more acidic/hydroxylated the support the more intense is the spillover effect.

Response: We are very grateful for the detailed comments from Reviewer 3 and also for bringing the recent review to our attention. The cited review presents a novel hypothesis for rationalizing the controversial hydrogen spillover effects over nonreducible metal oxides, in which the author proposed that active hydrogen atoms can effectively migrate on nonreducible metal oxides with aid of acidic surface OH groups via a H/OH exchange mechanism. We would like to point out first that this hypothesis is proposed purely based on the literature and still needs to be verified by experimental/theoretical evidence. For instance, the review states that “*the migration of D atoms over non-reducible oxide supports takes place by hopping from a hydroxyl to form a deuteroyl and not through D/OD pairs migration*”, while it is debatable whether H and D atoms would migrate on the nonreducible oxide surface via different mechanisms considering that the two isotopes have nearly identical chemical properties. Second, this hypothesis indicates the hydrogen spillover could take place on the nonreducible oxide surface only when the oxide surface has sufficient acidic OH groups such as zeolites. In contrast, what we found in our study is a potentially general and robust strategy for achieving hydrogen spillover on the nonreducible oxides regardless of whether acid sites or defects exist.

In References 9-11 mentioned by this reviewer, sodalite aluminosilicate zeolites with a high Al/Si ratio of ~1/1 were used as the support, and the abundant acidic surface OH groups on these supports were reported to play a key role in enabling hydrogen spillover on nonreducible oxides, which reflects, in turn, that effective hydrogen migration on nonreducible oxides with scarce acidic sites (e.g. the SiO₂ support used in our study) is almost impossible. In Reference 16, the hydrogen spillover phenomenon observed on the Pt-Co/SiO₂ catalyst was not analyzed by the authors, while it is noticeable that CO was formed as a predominate by-product during CO₂ methanation. We suspect that CO may act as a H-carrier to promote the spillover process, analogous to the case reported in our study. Regarding Nishiyama et al.’s paper (*J. Catal.* 2012), we agree with this reviewer the structure of the Pt-Sn/SiO₂ catalyst is very similar to that of the Pt-Fe/SiO₂ sample used in our study. However, these two catalysts showed distinct abilities for hydrogen spillover as reflected from the reduction temperature of the SnO₂/Fe₂O₃ species. Specifically, Nishiyama et al. showed that the presence of the core Pt nanoparticles significantly lowered the temperature required for complete reduction of the SnO₂ species residing on the outside surface of the SiO₂ shell (by ~100 K), which was attributed to the effect of hydrogen spillover. In contrast, the reduction

behavior of the Fe_2O_3 species in our Pt-Fe system was nearly unaffected by the spatially-separated Pt sites (1Pt@-10Fe@ SiO_2 vs. 10Fe@ SiO_2 in Fig. R7, adopted from Fig. 2D of the manuscript), indicating hydrogen spillover is inhibited on the SiO_2 support in our study. This difference between the Pt-Sn and Pt-Fe catalysts in hydrogen spillover is probably related to the surface properties of SiO_2 (e.g. the densities of acid sites and defects), which are sensitive to the synthesis method and thermal treatment of the catalysts. It is unfortunate that little information about the surface properties of the SiO_2 support can be found in Nishiyama et al.'s paper. At last, although the experiments in References 5 and 17 were conducted at a low pressure (1×10^{-5} mbar H_2), the conclusions obtained in these two studies are most likely still valid under realistic experimental conditions (at least) for nonreducible oxides with scarce acid sites, since the mean free path of H radicals on the oxide surface would be even shorter as the system pressure increases.

Taken together, effective hydrogen spillover on nonreducible metal oxides with a low density of acid sites or defects is still very challenging to the best of our knowledge, and our present study demonstrates with solid evidence that this can be achieved with aid of small oxygenate molecules. To address Reviewer 3's comment, the review article mentioned has been cited in the revised version of the manuscript as Reference 9, and the relevant statement in the introduction section has been revised as follows (highlighted in yellow, Page 2):

“Although some studies suggest that the presence of sufficient acidic surface OH groups or introducing defect sites into the oxide surface may promote the hydrogen transfer process⁹⁻¹¹, a general and robust strategy is still highly desired for achieving an effective hydrogen spillover on nonreducible oxides with a low density of acid sites or defects, which are widely used as catalyst supports in industry because of their excellent thermochemical stability...”

Fig. R7. H_2 -TPR profiles for the oxide precursors of 1Pt@ SiO_2 , 10Fe@ SiO_2 , and 1Pt@-Fe@ SiO_2 with a Pt-Fe/ SiO_2 catalyst (1wt% Pt, 10 wt% Fe) synthesized via a co-impregnation method as reference. Adopted from Fig. 2D of the manuscript.

Reviewer 3: 2. The HDO activity of 1Pt@SiO₂ and SiO₂ were similar, indicating the absence of hydrogen spillover. We agree with the authors that the postulated oxygenate species transporting hydrogen atoms may be considered. Nevertheless, close inspection of the results, shows that 1Pt@SiO₂, is a little more active (15.8%) than SiO₂ (11.2%), attributable to a small hydrogen spillover effect contribution. It is worth noting that compared to alumina [Teichner, Appl. catal., 1990] or zeolites [Benseraf, Appl. Catal., 2002] silica is less prone to hydrogen spillover effect. This could be the case here.

Response: Thank you for the kind reminder. We agree with Reviewer 3 that it is difficult to discern whether the slight difference of the HDO activity between 1Pt@SiO₂ and SiO₂ is contributed from a small hydrogen spillover effect. However, as addressed in the last response, the nearly identical reduction temperatures for the Fe₂O₃ species in 1Pt@-10Fe@SiO₂ and 10Fe@SiO₂ (Fig. R7) can rule out the possible hydrogen spillover effect. Furthermore, the similar HDO rates on 1Pt@-10Fe@SiO₂ and 10Fe@SiO₂ within a large H₂ partial pressure range (10-60 kPa) in the absence of cofeeding oxygenates (Fig. R8, adopted from Fig. S8 of the manuscript) also support the unlikeliness of hydrogen spillover on the nonreducible SiO₂ support used in our study. The later point was addressed in the original manuscript as (Page 8):

“...This similarity of catalytic performance between 1Pt@-10Fe@SiO₂ and 10Fe@SiO₂ in guaiacol HDO (verified within 10-60 kPa H₂ partial pressures, Fig. S8) thus provides compelling evidence for the unlikely hydrogen spillover between isolated Pt and Fe nanoparticles on the nonreducible SiO₂ support under the reaction conditions studied, which is consistent with previous reports^{5,17}.”

Fig. R8. Effect of H₂ pressure on the BTX formation rate (r_{BTX}) of guaiacol hydrodeoxygenation for 1Pt@-10Fe@/SiO₂ and 10Fe@SiO₂. Reaction condition: 450 °C, 0.5 kPa guaiacol, balanced by N₂, 0.25

g.s.mL⁻¹ space velocity. Dashed curves indicate trends. Adopted from Fig. S8 of the supporting information.

Reviewer 3: 3. When H₂ was supplied by the steam-reforming reaction (instead of gaseous H₂) and using 1Pt@-10Fe@SiO₂ as a catalyst, enhanced catalytic activity could be due to the presence of the water molecules added. This was shown in the gas phase hydrogenation of benzaldehyde with gold catalysts supported on alumina and in the presence of water vapor [Perret, *Catal. Comm.*, 2011]. Water dissociation produces hydroxyls which serve as bridges for the hydrogen spillover species transport on the support surface. This phenomenon has long been known [Benson, *J. Catal.*, 1966; Lenz, *J. Catal.*, 1988]. This may apply for the HDO of guaiacol. The H active species produced in the micropores on Pt could directly migrate to the macropores by the spillover effect with the help of H₂O. There is no need of oxygenate molecules to their transport.

If it is such the higher reaction rate (based on BTX production) of 1Pt@-10Fe@SiO₂ as compared to that of 10Fe@SiO₂ catalyst can be explained also by the enhanced spillover effect due to the water molecules. Similarly, the additional amounts of activated hydrogen do vary with the composition of the co-fed mixture (Figure 4) and the nature of the oxygenate molecule (Figure 7). In the same way the activity of 1Pt@-10Fe@SiO₂ decrease with the increase of the distance $\langle d_{Pt-Fe} \rangle$ between Pt and iron (Figure 5). This is an expected process for hydrogen spillover whatever the nature of its source.

Response: When we examined the effects of C₁-C₃ oxygenate additives on the HDO activity, we were indeed aware of the possible interference by H₂O, same as Reviewer 3. The experiments shown in Fig. 7a of the manuscript (attached below as Fig. R9) were, in fact, conducted without cofeeding H₂O. This allowed us to attribute the observed enhancement of the HDO activity completely to the introduction of the oxygenate additives. We have emphasized this experimental detail in the revised manuscript as follows (highlighted in yellow, Page 13):

“Therefore, we further examined the effects of these oxygenate additives on the rates of guaiacol HDO over 1Pt@-10Fe@SiO₂, and these experiments were conducted without cofeeding H₂O in order to avoid the possible formation of surface OH groups on the SiO₂ support that may promote the hydrogen spillover⁹.”

Fig. R9. The formation rate of BTX (r_{BTX}) as a function of H_2 partial pressure with or without 0.5 kPa oxygenate additive for guaiacol hydrodeoxygenation on $1\text{Pt}@-10\text{Fe}@\text{SiO}_2$. Reaction condition: $450\text{ }^\circ\text{C}$, 0.5 kPa guaiacol, balanced by N_2 , methanol and ethanol used as the precursors of formaldehyde and acetaldehyde, respectively. Solid curves indicate trends. Adopted from Fig. 7a of the manuscript.

Reviewer 3: 4. From the above considerations it would be of interest to perform the HDO reaction over $1\text{Pt}@\text{SiO}_2$ catalyst using $\text{CH}_3\text{OH}+\text{H}_2\text{O}$ co-feeding mixture and compare to reaction using H_2 co-feeding.

Response: Thank you for this interesting suggestion. We have conducted the experiment which shows that the conversion and selectivity of guaiacol HDO on $1\text{Pt}@\text{SiO}_2$ at 50 kPa H_2 are very close to those obtained when the mixture of CH_3OH (0.5 kPa) and H_2O (0.5 kPa) were cofed instead of H_2 (Fig. R10). Since the effective H_2 pressure generated from this $\text{CH}_3\text{OH}+\text{H}_2\text{O}$ mixture was as low as 1.3 kPa, it is indicated the CH_3OH additive can also promote the guaiacol conversion on the active sites of SiO_2 via enabling hydrogen spillover, which is similar to that found for the Pt-Fe system. It is worth mentioning that the contribution from the SiO_2 support in the Pt-Fe system is minor for guaiacol HDO, because the SiO_2 support exhibits much lower hydrodeoxygenation activity than the Fe nanoparticles. Fig. R10 has been added in the supporting information as Fig. S9, and a discussion about these data has been included in the revised manuscript as follows (Page 8):

“An activity enhancement of the guaiacol conversion was also observed for the $1\text{Pt}@\text{SiO}_2$ catalyst (Fig. S9), despite $1\text{Pt}@\text{SiO}_2$ was much less active than $1\text{Pt}@-10\text{Fe}@\text{SiO}_2$ and showed a high selectivity to PAC (Table 1). It is thus suggested that this promoting effect brought forth by the co-feeding of CH_3OH and H_2O appears to be general for the catalysts with spatially-separated H_2 -activation and hydrodeoxygenation sites.”

Fig. R10. Catalytic performance of gas-phase guaiacol hydrodeoxygenation over 1Pt@SiO₂. Reaction condition: 450 °C, 0.5 kPa guaiacol, balanced by N₂, 0.25 g·s·mL⁻¹ space velocity. ^a Gaseous H₂ (50 kPa) was directly fed into the reactor. ^b CH₃OH (0.5 kPa) and H₂O (0.5 kPa) were cofed into the reactor to generate H₂ in situ via methanol steam reforming (corresponding to an effective H₂ pressure of 1.3 kPa). Here, BTX is denoted for the sum of benzene, toluene, and xylene, while PAC is denoted for the sum of phenol, anisole, and cresol.

Reviewer 3: 5. If the oxygenate transporting species exist then they could be evidenced by radical chain inhibitors, e. g. NO.

Response: Following this suggestion from Reviewer 3, we conducted an additional experiment by cofeeding NO during guaiacol HDO in order to verify the existence of oxygenate transporting species. As shown in Fig. R11, the BTX formation rate on 1Pt@-10Fe@SiO₂ was significantly suppressed in the presence of 3.5 kPa NO with nearly full recovery after removing NO, providing evidence for the radical nature of the oxygenate transporting species in hydrogen spillover. Fig. R11 has been added in the supporting information as Fig. S11, and a discussion about these new data has been included in the revised manuscript as follows (Page 12):

“The radical nature of such oxygenate transporting species during guaiacol HDO on 1Pt@-10Fe@SiO₂ was further confirmed by the observation of a significant inhibiting effect of NO, a radical chain inhibitor, on the BTX formation rate (Fig. S11).”

Fig. R11. Effect of NO cofeeding on the formation rate of BTX (r_{BTX}) over 1Pt@-10Fe@SiO₂. Reaction condition: 450 °C, 0.5 kPa guaiacol, 20 kPa H₂, 0.5 kPa CH₃OH, and cofeeding with or without 3.5 kPa NO, balanced by N₂, 0.25 g·s·mL⁻¹ space velocity.

REVIEWER COMMENTS

Reviewer #1 (Remarks to the Author):

I believe that the authors carefully addressed all the issues raised by the reviewers. The manuscript can be accepted as it is.

Reviewer #3 (Remarks to the Author):

1. First of all I appreciate the conscientiousness of the authors who performed the experiments suggested by the referees. This is the case of the use of NO as a probe molecule confirming the existence of radical transient species in the HDO of guaiacol when using CH₃OH+H₂ as co-feeding (Figure S11). However, the reaction paths involved may be different from that proposed by the authors (see below).

2. Frankly speaking the results presented in Figure S11 do not correspond to that I demanded, that is the effect of NO on methanol steam reforming co-feeding, namely to compare guaiacol/CH₃OH+H₂O and guaiacol/CH₃OH+H₂O/NO reactive atmospheres. I conclude that the claiming of the authors about the existence of a radical mechanism for guaiacol/CH₃OH+H₂O mixture is not proved. It holds for the guaiacol/CH₃OH+H₂ catalytic system only and probably also when other oxygenates were used instead of methanol.

Water is an oxidant which modifies the redox potential of the catalyst surface and therefore is expected to modify the reaction paths and rates. The thermodynamics of the two catalytic systems are different.

3. The authors rebuttal of hydrogen spillover effect in the case of metal catalysts supported on nonreducible oxides may be due to a too rapid reading of the related paper (Catal. Rev. 2020). In short, if I agree that hydrogen spillover effect would be limited in the case of the silica support (probably too much dehydroxylated), it should be promoted in the presence of water. The effect of water on hydrogen spillover is well documented.

Certain results may be interpreted in this way:

i) In Figure S9 1Pt@SiO₂ catalyst exhibits an enhanced activity when cofed with CH₃OH+H₂O as compared with CH₃OH+H₂ mixture: the H atoms formed on Pt nanoparticles encapsulated in the micropores of silica spilt over to the support surface where the HDO took place. The spillover was facilitated by additional OH groups stemming from the dissociation of water. These OH groups served as bridges along the reaction path from Pt to silica active sites.

ii) In Figure S10 it is seen that the BTX formation rate was the highest with 1Pt@10Fe@SiO₂ as compared to 10Fe@SiO₂ when CH₃OH+H₂O cofeeding was used. This may be ascribed to the higher efficiency of Pt as compared to Fe in the steam reforming of methanol. Higher amounts of H species formed on Pt which spilt over to Fe where they reacted with guaiacol.

4. The co-feedings in Figures 4 and 5 (CH₃OH+H₂O) are different from that in Figures 7 and S11 (oxygenates+H₂). The rates cannot be directly compared. Figures 4 and 5 not joint to the present report.

5. Figure 7A shows the influence of the nature of the oxygenate reactant on the formation rate of BTX. DFT calculations reported in Figure 7B seem to corroborate the existence of hydrogenated radical

carbonyl species as H transporting agents from Pt to Fe. However, another interpretation exists. In effect, methanol and ethanol are well known as in-situ H₂ supplier in the HDO of guaiacol, avoiding the costly use of external H₂. In their decomposition to CO+H₂ mixture, formaldehyde or acetaldehyde are intermediately formed. These species are able to spillover from Pt to Fe where they could deoxygenate guaiacol:

$RCH_2OH \Rightarrow [RCHO] + H_2 \Rightarrow CO + 2H_2$ R = H or CH₃

guaiacol + [RCH=O] \Rightarrow catechol/anisole + ...

This is a hydrogen transfer reaction. It also holds for acetone or methylformate molecules.

The NO experiments indicated that the overall reaction process was radical in nature. Precursors of formaldehyde or acetaldehyde, catechol or anisole could be radical species. There is no need to consider the existence of hydrogenated carbonyl radicals.

If it is such what would be the role of H₂ in the oxygenate+H₂ mixtures. May it be just an additional H atoms supplier?

7. Generally speaking, the HDO conversions reported (89- 96%) are too high and may hinder the real surface state of the catalysts. Reactions kinetics at low conversions would have been helpful in the comprehending of the reaction.

8. To conclude, the hierarchical PtFeSiO₂ catalysts studied are of interest from the fundamental point of view. However, we surmise a misinterpretation of the results obtained. This would come from not enough attention paid to the chemical and thermodynamics properties of the reactions when different hydrogenating agents [H₂, oxygenate+H₂, oxygenate+H₂O] were used.

We think that that guaiacol HDO is a completely different reaction according to the hydrogenating agent: oxygenates+H₂ versus in oxygenate +H₂O. The role of the individual molecules in the reactive atmosphere were not clarified. The radical nature of the HDO with guaiacol/CH₃OH+H₂ catalytic system not with guaiacol/CH₃OH+H₂O catalytic system. Hydrogen spillover effect cannot be excluded considered.

The manuscript needs a strong revision.

Response to Reviewers

The point-by-point response to the two reviewers is addressed below. We are very grateful for the reviewers' time and valuable comments and have tried our best to revise the manuscript accordingly.

Response to Reviewer 1:

Reviewer 1: *I believe that the authors carefully addressed all the issues raised by the reviewers. The manuscript can be accepted as it is.*

Response: We appreciate this reviewer's acknowledgement and recommendation on our revised manuscript.

Response to Reviewer 3:

Reviewer 3: *1. First of all, I appreciate the conscientiousness of the authors who performed the experiments suggested by the referees. This is the case of the use of NO as a probe molecule confirming the existence of radical transient species in the HDO of guaiacol when using CH₃OH+H₂ as co-feeding (Figure S11). However, the reaction paths involved may be different from that proposed by the authors (see below).*

Response: We thank Reviewer 3 for the acknowledgement on our effort in elucidating the oxygenate-assisted hydrogen spillover mechanism and his/her constructive comments provided below. On one hand, we would like to clarify that the mentioned experiments were designed to specifically confirm whether the radical transient species exist in the HDO of guaiacol when CH₃OH and H₂ were cofed. We were aware of the possible complex effects of H₂O emphasized by this reviewer and purposely designed the experiment to confirm without ambiguity that small oxygenate molecules can enable an effective hydrogen spillover on the inert SiO₂ surface by themselves in the absence of H₂O. This would confer a potentially more general and robust strategy for

achieving hydrogen spillover on the nonreducible oxides regardless of whether acid sites or defects exist. On the other hand, we fully agree with this reviewer that we should have made a more clear elucidation on the potential role of H₂O involved in the HDO of guaiacol with cofeeding CH₃OH and H₂O. As shown in the following responses, we have conducted additional experiments to resolve this reviewer's remaining concern. We hope this newly revised version of our manuscript provides a more rigorous and convincing demonstration on the oxygenate-assisted hydrogen spillover mechanism.

Reviewer 3: 2. Frankly speaking the results presented in Figure S11 do not correspond to that I demanded, that is the effect of NO on methanol steam reforming co-feeding, namely to compare guaiacol/CH₃OH+H₂O and guaiacol/CH₃OH+H₂O/NO reactive atmospheres. I conclude that the claiming of the authors about the existence of a radical mechanism for guaiacol/CH₃OH+H₂O mixture is not proved. It holds for the guaiacol/CH₃OH+H₂ catalytic system only and probably also when other oxygenates were used instead of methanol.

Water is an oxidant which modifies the redox potential of the catalyst surface and therefore is expected to modify the reaction paths and rates. The thermodynamics of the two catalytic systems are different.

Response: We thank the reviewer for reminding us of this critical issue. To address the reviewer's concern, we have conducted the suggested experiments and results are shown in Fig. R1. Similar to the case with CH₃OH+H₂ cofeeding, the addition of NO reversibly inhibited the HDO rate of guaiacol when a CH₃OH+H₂O mixture was used to generate H₂ in situ via methanol steam reforming. This new result confirms the existence of a radical mechanism for the guaiacol/CH₃OH+H₂O catalytic system. We should note that the 1Pt@-10Fe@SiO₂ catalyst slightly deactivated with time-on-stream during cofeeding CH₃OH and H₂O, which is due to the oxidation of metallic Fe by H₂O under low H₂ pressures as reported in our recent study (*ACS Catal.* 2020, 10, 7884-7893). Nevertheless, the HDO pathway on the Fe surface seems to be unaffected by cofeeding H₂O, evidenced by the similar product distributions obtained in H₂ and

CH₃OH+H₂O atmospheres (compared at ~30% guaiacol conversion, Fig. R2).

Fig. R1. Effect of NO cofeeding on the formation rate of BTX (r_{BTX}) over 1Pt@10Fe@SiO₂ with H₂ generated in situ from a mixture of CH₃OH and H₂O. Reaction condition: 450 °C, 0.5 kPa guaiacol, 0.5 kPa CH₃OH, 0.5 kPa H₂O, and cofeeding with or without 3.5 kPa NO, balanced by N₂, 0.25 g·s·mL⁻¹ space velocity.

Fig. R2. Catalytic performance of gas-phase guaiacol hydrodeoxygenation over 1Pt@10Fe@SiO₂. Reaction condition: 450 °C, 0.5 kPa guaiacol, balanced by N₂, 1.25 g·s·mL⁻¹ space velocity. ^a Gaseous H₂ (20 kPa) was directly fed into the reactor. ^b CH₃OH (0.5 kPa) and H₂O (0.5 kPa) were cofed into the reactor to generate H₂ in situ via methanol steam reforming (corresponding to an effective H₂ pressure of 1.3 kPa). Here, BTX is denoted for the sum of benzene, toluene, and xylene, while PAC is denoted for the sum of phenol, anisole, and cresol. Note: The guaiacol conversion obtained with cofeeding CH₃OH and H₂O (i.e. ~1.3 kPa H₂) was nearly identical to the case with directly feeding H₂ of 20 kPa, reflective of an enhanced HDO activity in the presence of the CH₃OH-H₂O mixture.

In order to examine the role of H₂O itself on hydrogen spillover, we have further compared the HDO rates of guaiacol on the 1Pt@-10Fe@SiO₂ catalyst in various reactive atmospheres. As shown in Fig. R3, the addition of H₂O (0.5 kPa) did not apparently affect the initial HDO rate under 10 kPa H₂, whereas the HDO rate was nearly tripled when CH₃OH (0.5 kPa) was cofed instead of H₂O. These results indicate that, different from CH₃OH and other small oxygenates, H₂O does not promote hydrogen spillover on the SiO₂ support (at least) under the condition of guaiacol HDO, probably because of the low H₂O pressure used in this study which limits its potential effects.

Fig. R3. Effect of additive on the formation rate of BTX (r_{BTX}) over 1Pt@-10Fe@SiO₂. Reaction condition: 450 °C, 0.5 kPa guaiacol, 10 kPa H₂, 0.5 kPa H₂O or CH₃OH cofed, balanced by N₂, 0.25 g·s·mL⁻¹ space velocity.

Based on the above discussions, Fig. R1 has been included in the supporting information as Fig. S13, and a relevant content in the manuscript has been revised as follows (highlighted in yellow, Page 12).

“The radical nature of such oxygenate transporting species during guaiacol HDO on 1Pt@-10Fe@SiO₂ was further confirmed by the observation of a significant inhibiting effect of NO, a radical chain inhibitor, on the BTX formation rate **no matter whether H₂ was directly fed or generated in situ from a mixture of CH₃OH and H₂O (Figs. S12 and S13).**”

Fig. R2 has been added in the supporting information as Fig. S9, together with a sentence included in the revised manuscript as (highlighted in yellow, Page 8):

“This r_{BTX} value achieved with co-feeding CH_3OH and H_2O was even higher than that obtained with co-feeding 50 kPa H_2 ($0.84 \text{ mol mol}_{\text{Fe}}^{-1} \text{ h}^{-1}$, Fig. 4). Such a promoting effect of the $\text{CH}_3\text{OH-H}_2\text{O}$ mixture was further confirmed for guaiacol HDO at lower conversions (i.e. $\sim 30\%$, Fig. S9), in which PAC existed as the main products.”

Fig. R3 has also been included in the supporting information (denoted as Fig. S14), and the corresponding content added in the revised manuscript is (highlighted in yellow, Page 12):

“The radical nature of such oxygenate transporting species during guaiacol HDO on 1Pt@-10Fe@SiO_2 was further confirmed by the observation of a significant inhibiting effect of NO , a radical chain inhibitor, on the BTX formation rate no matter whether H_2 was directly fed or generated in situ from a mixture of CH_3OH and H_2O (Figs. S12 and S13). It is also worth noting that the possible role of H_2O in promoting the H transfer over SiO_2^9 was excluded based on the negligible effect of H_2O addition on the HDO activity of 1Pt@-10Fe@SiO_2 at 10 kPa H_2 (Fig. S14).”

Reviewer 3: 3. The authors' rebuttal of hydrogen spillover effect in the case of metal catalysts supported on nonreducible oxides may be due to a too rapid reading of the related paper (Catal. Rev. 2020). In short, if I agree that hydrogen spillover effect would be limited in the case of the silica support (probably too much dehydroxylated), it should be promoted in the presence of water. The effect of water on hydrogen spillover is well documented.

Certain results may be interpreted in this way:

i) In Figure S9, 1Pt@SiO_2 catalyst exhibits an enhanced activity when cofed with $\text{CH}_3\text{OH+H}_2\text{O}$ as compared with $\text{CH}_3\text{OH+H}_2$ mixture: the H atoms formed on Pt nanoparticles encapsulated in the micropores of silica spilt over to the support surface where the HDO took place. The spillover was facilitated by additional OH groups stemming from the dissociation of water. These OH groups served as bridges along the reaction path from Pt to silica active sites.

ii) In Figure S10 it is seen that the BTX formation rate was the highest with 1Pt@10Fe@SiO₂ as compared to 10Fe@SiO₂ when CH₃OH+H₂O cofeeding was used. This may be ascribed to the higher efficiency of Pt as compared to Fe in the steam reforming of methanol. Higher amounts of H species formed on Pt which spilt over to Fe where they reacted with guaiacol.

Response: We would like to thank the reviewer again for bringing the critical review to our attention. We read through the review article carefully and were very aware of the possible effects of H₂O on hydrogen spillover. As elucidated in our previous response, our latest data show that similar HDO rates were obtained for the H₂ and H₂+H₂O conditions (Fig. R3), ruling out the contribution of H₂O in promoting hydrogen spillover on 1Pt@-10Fe@SiO₂ (at least) under the reaction condition of guaiacol HDO. We surmise that a combination of the high reaction temperature (i.e. 450 °C) and the low pressure of cofed H₂O (i.e. 0.5 kPa) used in our studies may have kept the SiO₂ surface from forming a high degree of hydroxylation, which, in fact, provides us the opportunity to unveil the oxygenate-assisted hydrogen spillover pathway on the inert SiO₂ support.

Reviewer 3: 4. The co-feedings in Figures 4 and 5 (CH₃OH+H₂O) are different from that in Figures 7 and S11 (oxygenates+H₂). The rates cannot be directly compared. Figures 4 and 5 are not joint to the present report.

Response: We appreciate this suggestion from Reviewer 3, while we believe Figures 4 and 5 are essential for the present report. First, the oxygenate-assisted hydrogen spillover on an inert SiO₂ support was discovered during our initial attempt to generate equivalent H₂ in situ via methanol steam reforming for the catalytic HDO of guaiacol, and the data in Figures 4 and 5 provide key evidence for the existence of hydrogen spillover enabled by the CH₃OH+H₂O mixture, which conferred the basis for us to further investigate the role of small oxygenate molecules by feeding the oxygenate additives solely as shown in Figures 7 and S11. In our opinion, Figures 4 and 5 are

logically joint to the present report. Second, Figures 4 and 5 exhibit the advantage of using CH₃OH+H₂O mixtures as an alternative H₂ source for HDO reactions, which not only avoids the direct use of expensive H₂ gas, but also promotes hydrogen spillover with aid of formaldehyde generated in situ from methanol. Considering such attractive advantages for catalytic hydrodeoxygenation reactions, it is worth keeping Figures 4 and 5 in the present report as well.

Reviewer 3: 5. Figure 7A shows the influence of the nature of the oxygenate reactant on the formation rate of BTX. DFT calculations reported in Figure 7B seem to corroborate the existence of hydrogenated radical carbonyl species as H transporting agents from Pt to Fe. However, another interpretation exists.

In effect, methanol and ethanol are well known as in-situ H₂ supplier in the HDO of guaiacol, avoiding the costly use of external H₂. In their decomposition to CO+H₂ mixture, formaldehyde or acetaldehyde are intermediately formed. These species are able to spillover from Pt to Fe where they could deoxygenate guaiacol:

This is a hydrogen transfer reaction. It also holds for acetone or methylformate molecules.

The NO experiments indicated that the overall reaction process was radical in nature. Precursors of formaldehyde or acetaldehyde, catechol or anisole could be radical species. There is no need to consider the existence of hydrogenated carbonyl radicals. If it is such what would be the role of H₂ in the oxygenate+H₂ mixtures. May it be just an additional H atoms supplier?

Response: We are afraid that the reviewer's assumption about the deoxygenation of guaiacol by carbonyl compounds (e.g., formaldehyde and acetaldehyde) might be unlikely to happen. As shown in Fig. 3 of the manuscript (attached here as Fig. R4), the deoxygenation of guaiacol to BTX requires a hydrogen supplier to remove the oxygen atoms of guaiacol in terms of H₂O. Apparently, it is much more difficult for those

carbonyl compound intermediates (formed from alcohol dehydrogenation) to offer H-atoms compared with their alcohol parents. In other words, if a hydrogen transfer reaction occurs, the preferred hydrogen supplier would be the alcohols, instead of the carbonyl compounds or esters. However, as addressed in our response to the reviewers from the previous round of revisions, it would be expected to see similar initial HDO rates for the Pt-Fe and Fe catalysts with cofeeding methanol if transfer hydrogenation does exist, since methanol can access the Fe surface in both cases. The apparently higher hydrodeoxygenation rate on Pt-Fe than Fe with methanol (Fig. R5, adopted from Fig. S11 of the Supplementary Information) thus excludes the contribution of hydrogen transfer reactions, which supports, in turn, the existence of hydrogen spillover brought forth by the oxygenate additives.

The formation of hydrogenated carbonyl radicals is surmised to explain how hydrogen spillover on the inert SiO₂ support is enabled by the oxygenate additives. As discussed in the manuscript, this hypothesis is consistent with our DFT calculations and NO-cofeeding experiments. We would also like to point out that, in those experiments with the oxygenate+H₂ mixtures (e.g. Fig. 7 of the manuscript), the H₂ pressure used (10-60 kPa) was much higher than that of the oxygenate additive (0.5 kPa), making the H₂ amount generated in situ from the alkanol additives negligible. Therefore, for the oxygenate+H₂ mixtures, the oxygenate additive and H₂ act as the H-atom carrier (or the carrier precursor) and the H-atom supplier, respectively.

Fig. R4. Reaction pathways of guaiacol hydrodeoxygenation. Only relevant products observed in this study are included here. Adopted from Fig. 3 of the manuscript.

Fig. R5. Comparison of the BTX formation rate (r_{BTX}) in guaiacol hydrodeoxygenation between 10Fe@/SiO₂ and 1Pt@-10Fe@SiO₂ when CH₃OH and H₂O were cofed instead of H₂. Reaction condition: 450 °C, 0.5 kPa guaiacol, 0.5 kPa CH₃OH, 0.5 kPa H₂O, balanced by N₂, 0.25 g.s.mL⁻¹ space velocity. Adopted from Fig. S11 of the supporting information.

Reviewer 3: 7. Generally speaking, the HDO conversions reported (89- 96%) are too high and may hinder the real surface state of the catalysts. Reactions kinetics at low conversions would have been helpful in the comprehending of the reaction.

Response: We thank the reviewer for this constructive suggestion. In the present study, high HDO conversions had to be performed for meaningfully achieving high BTX selectivities, because these target aromatic products are generated from guaiacol via a series of sequential reactions (Fig. R4). On the other hand, the initial pressure of guaiacol in these HDO conversions was kept as low as 0.5 kPa, which should restrain the possible effects of high HDO conversions on the real surface state of the catalysts. In order to examine our hypothesis, we have compared the catalytic activities of 1Pt@-10Fe@SiO₂ with and without cofeeding CH₃OH+H₂O mixtures at low HDO conversions of ~30%. As shown in Fig. R2, the guaiacol conversion obtained with cofeeding CH₃OH and H₂O (i.e. ~1.3 kPa H₂) was nearly identical to the case with directly feeding H₂ of 20 kPa, reflective of an enhanced HDO activity in the presence of the CH₃OH-H₂O mixture. It is thus indicated that the oxygenate-assisted hydrogen spillover mechanism is valid regardless of the HDO conversion.

Reviewer 3: 8. To conclude, the hierarchical PtFeSiO₂ catalysts studied are of interest from the fundamental point of view. However, we surmise a misinterpretation of the results obtained. This would come from not enough attention paid to the chemical and thermodynamics properties of the reactions when different hydrogenating agents [H₂, oxygenate+H₂, oxygenate+H₂O] were used.

We think that that guaiacol HDO is a completely different reaction according to the hydrogenating agent: oxygenates+H₂ versus in oxygenate+H₂O. The role of the individual molecules in the reactive atmosphere were not clarified. The radical nature of the HDO with guaiacol/CH₃OH+H₂ catalytic system not with guaiacol/CH₃OH+H₂O catalytic system. Hydrogen spillover effect cannot be excluded considered.

The manuscript needs a strong revision.

Response: We thank the reviewer again for helping us improve the manuscript. As described in the above responses, our new experiments indicate the chemistry of guaiacol HDO is not inherently altered when the reactive atmosphere is changed from oxygenate+H₂ to oxygenate+H₂O. The cofed H₂O merely acts as a reactant of the steam reforming of oxygenate to generate H₂ in situ, and the oxygenate additives are H-atom carriers or carrier precursors that can enable hydrogen spillover on the inert SiO₂ support under the condition of guaiacol HDO. In order to highlight our findings, we included an illustration of the oxygenate-assisted hydrogen spillover mechanism as Fig. 7C in the revised manuscript (attached here as Fig. R6).

Fig. R6. An illustration of the proposed oxygenate-assisted hydrogen spillover mechanism on the inert SiO₂ support.

REVIEWERS' COMMENTS

Reviewer #3 (Remarks to the Author):

I again appreciate the conscientiousness of the authors who performed the experiments suggested.

1. These experiments clearly show that H₂O does not play a role in the HDO of guaiacol when using the CH₃OH+H₂O mixture as a hydrogen supplying agent. It does not enhance the classical hydrogen spillover under the conditions used, as I personally expected. Also the transport of hydrogen by another way (oxygenate molecules or radicals) is the right explanation.

2. Aldehydes (alkylformates) are good reducing agents (uses chemistry) due to the high reactivity of the H atom of the CH=O function (a hydride source). For methanol the reaction would as follows:

In such a case there is no need of an external source of hydrogen. It is the reason why I suggested (previous review report) that they could participate to the HDO of guaiacol after spilling over from Pt to Fe.

3. In the absence of gaseous hydrogen the reaction did not take place (Figure 7A). This may indicate that, in this case, aldehydes were not the right H transfer agent, which invalidate my suggestion. Nevertheless, this may also be due to active sites blocking. In effect, as currently observed, HDO catalysts deactivate with time on stream due to poisoning of the active sites by adsorbed oxygenate species. Gaseous hydrogen is a good solid surface cleaning agent also, in its absence, the reaction may be blocked. Could the authors comment this point, notably on the deactivation of their catalysts ?

4. It would have been better to report the demonstration of the radical nature of guaiacol HDO in the core of the manuscript, not as supplementary information, all the more it is an enlargement of the knowledge on this reaction.

In conclusion, since the transport of hydrogen by organic molecules would be an original scientific result, there is a need to gather as much evidence as possible. The manuscript could be considered for publication with minor revision.

Response to Reviewers

The point-by-point response to Reviewer 3 is addressed below. We are very grateful for the time and effort of the reviewer in promoting our work and we have revised the manuscript again based on the latest suggestions.

Reviewer 3: *I again appreciate the conscientiousness of the authors who performed the experiments suggested.*

1. These experiments clearly show that H₂O does not play a role in the HDO of guaiacol when using the CH₃OH+H₂O mixture as a hydrogen supplying agent. It does not enhance the classical hydrogen spillover under the conditions used, as I personally expected. Also the transport of hydrogen by another way (oxygenate molecules or radicals) is the right explanation.

Response: We thank the reviewer again for the constructive comments provided previously that enable us to strengthen our conclusions on the oxygenate-assisted hydrogen spillover mechanism.

Reviewer 3: *2. Aldehydes (alkylformates) are good reducing agents (oses chemistry) due to the high reactivity of the H atom of the CH=O function (a hydride source). For methanol the reaction would as follows:*

In such a case there is no need of an external source of hydrogen. It is the reason why I suggested (previous review report) that they could participate to the HDO of guaiacol after spilling over from Pt to Fe.

Response: We thank the reviewer for reminding us of this possibility. As mentioned in our previous response, the H₂ pressure used in those experiments with the oxygenate+H₂ mixtures (e.g., 10-60 kPa in Fig. 7 of the manuscript) was much higher

than that of the oxygenate additive (0.5 kPa), making the H₂ amount generated in situ from the oxygenate additives negligible. Therefore, the promoting effect of the oxygenate additives on the HDO activity can be primarily attributed to the effective hydrogen spillover enabled by the oxygenate additives, as expected by the reviewer.

Reviewer 3: 3. In the absence of gaseous hydrogen the reaction did not take place (Figure 7A). This may indicate that, in this case, aldehydes were not the right H transfer agent, which invalidate ma suggestion. Nevertheless, this may also be due to active sites blocking. In effect, as currently observed, HDO catalysts deactivate with time on stream due to poisoning of the active sites by adsorbed oxygenate species. Gaseous hydrogen is a good solid surface cleaning agent also, in its absence, the reaction may be blocked. Could the authors comment this point, notably on the deactivation of their catalysts?

Response: We fully agree with the reviewer's comment and have discussed the deactivation issue in the revised manuscript as follows (highlighted in yellow, Page 15):

“... It is also worth noting that the possible role of H₂O in promoting the H transfer over SiO₂⁹ was excluded based on the negligible effect of H₂O addition on the HDO activity of 1Pt@-10Fe@SiO₂ at 10 kPa H₂ (Fig. S14). On the other hand, a slight deactivation of the 1Pt@-10Fe@SiO₂ catalyst was observed with time-on-stream when cofeeding CH₃OH and H₂O (Fig. S13), which is mainly due to the oxidation of metallic Fe by H₂O or active site blocking by adsorbed oxygenate species^{27,28}. This suggests that a sufficient H₂ partial pressure, which is either cofed or formed in situ, is requisite for preventing the Fe-based catalysts from deactivation during HDO.”

Reviewer 3: 4. It would have been better to report the demonstration of the radical nature of guaiacol HDO in the core of the manuscript, not as supplementary information, allthemore it is an enlargement of the knowledge on this reaction.

Response: We thank Reviewer 3 for the kind suggestion. However, we would like to keep the organization of the content in the manuscript as it is, because we don't want to dilute the focus of the paper, namely the oxygenate-assisted hydrogen spillover on inert oxides.

Reviewer 3: *In conclusion, since the transport of hydrogen by organic molecules would be an original scientific result, there is a need to gather as much evidence as possible. The manuscript could be considered for publication with minor revision.*

Response: We greatly appreciate the reviewer's suggestion and recommendation.